# Impairment of cocaine-mediated behaviours in mice by clinically relevant Ras-ERK inhibitors

**Alessandro Papale**[1,2†], **Ilaria Maria Morella**[1,2†], **Marzia Tina Indrigo**[3], **Rick Eugene Bernardi**[4,5,6], **Livia Marrone**[7], **Francesca Marchisella**[7], **Andrea Brancale**[8], **Rainer Spanagel**[4,5,6], **Riccardo Brambilla**[1,2*], **Stefania Fasano**[1,2*]

[1]Neuroscience and Mental Health Research Institute, Cardiff University, Cardiff, United Kingdom; [2]School of Biosciences, Cardiff University, Cardiff, United Kingdom; [3]IRCCS-Istituto di Ricerche Farmacologiche Mario Negri, Milan, Italy; [4]Institute of Psychopharmacology, Heidelberg University, Heidelberg, Germany; [5]Central Institute of Mental Health, Heidelberg University, Heidelberg, Germany; [6]Medical Faculty Mannheim, Heidelberg University, Heidelberg, Germany; [7]Institute of Experimental Neurology, Division of Neuroscience, IRCCS-San Raffaele Scientific Institute, Milan, Italy; [8]School of Pharmacy and Pharmaceutical Sciences, Cardiff University, Cardiff, United Kingdom

**\*For correspondence:**
brambillar@cardiff.ac.uk (RB);
fasanos@cardiff.ac.uk (SF)

†These authors contributed equally to this work

**Competing interests:** The authors declare that no competing interests exist.

**Abstract** Ras-ERK signalling in the brain plays a central role in drug addiction. However, to date, no clinically relevant inhibitor of this cascade has been tested in experimental models of addiction, a necessary step toward clinical trials. We designed two new cell-penetrating peptides - RB1 and RB3 - that penetrate the brain and, in the micromolar range, inhibit phosphorylation of ERK, histone H3 and S6 ribosomal protein in striatal slices. Furthermore, a screening of small therapeutics currently in clinical trials for cancer therapy revealed PD325901 as a brain-penetrating drug that blocks ERK signalling in the nanomolar range. All three compounds have an inhibitory effect on cocaine-induced ERK activation and reward in mice. In particular, PD325901 persistently blocks cocaine-induced place preference and accelerates extinction following cocaine self-administration. Thus, clinically relevant, systemically administered drugs that attenuate Ras-ERK signalling in the brain may be valuable tools for the treatment of cocaine addiction.

## Introduction

Exposure to drugs of abuse such as cocaine produces intense and long-lasting memories that are critical in the transition from recreational drug-taking to compulsive and uncontrolled drug use (*Everitt, 2014*). Several lines of evidence have demonstrated that the experience of drugs of abuse depends on learned associations between drug-paired cues and the rewarding effects of these drugs (*Sanchis-Segura and Spanagel, 2006*). Thus, preventing relapse following drug abstinence is impeded by the ability of drug-associated cues to persistently elicit drug-seeking behaviours (*Belin et al., 2013*). As a consequence, conditioned drug cues may induce drug seeking and relapse behaviour. In the brain, cellular mechanisms and signalling pathways involved in normal learning and memory processes are usurped by addictive drugs; hence, memories encoding drug-paired cues resist extinction and contribute to high rates of relapse (*Nestler, 2004*; *Miller and Marshall, 2005*; *Torregrossa et al., 2011*; *Itzhak et al., 2014*).

In particular, Ras-ERK signalling has been implicated in both the acute and long-term effects of cocaine using different experimental paradigms that mimic drug addiction in humans (*Valjent et al., 2000*; *Lu et al., 2005*; *Valjent et al., 2006a*, *2006b*; *Lu et al., 2006*; *Ferguson et al., 2006*; *Girault et al., 2007*; *Fasano et al., 2009*; *Fasano and Brambilla, 2011*; *Pascoli et al., 2011*, *2012*, *2014*; *Cahill et al., 2014*; *Garcia-Pardo et al., 2016*). The use of pharmacological inhibitors of MEK1/2, the kinases upstream of ERK1/2, has been essential in defining the role of ERK cascade in lasting experience- and drug-dependent alterations in behavioural plasticity. Until recently, the only available blood-brain barrier penetrating MEK inhibitor was SL327, but this drug has never been used in humans, most likely due to toxicity issues and to the relatively low IC50 (0.18/0.22 $\mu$M toward MEK1 and MEK2, respectively) (*Atkins et al., 1998*; *Scherle et al., 2000*). Another MEK inhibitor unable to pass the blood-brain barrier but still widely used via direct injection in the brain is U0126 (*Favata et al., 1998*). Regardless, it has been shown that MEK inhibition prevents conditioned place preference (CPP) to cocaine and amphetamine (*Valjent et al., 2000*; *Gerdjikov et al., 2004*; *Miller and Marshall, 2005*; *Valjent et al., 2006a*). Other studies have identified a key role for ERK in drug reinforcement using the self-administration paradigm. Cocaine self-administration has been demonstrated to increase ERK phosphorylation relative to saline controls in striatal regions that mediate reinforcement, including the nucleus accumbens shell (*Edwards et al., 2007*; *Miszkiel et al., 2014*). In addition, drug-seeking during exposure to cocaine-associated cues/contexts following the acquisition of self-administration has also been demonstrated to be ERK-dependent, although its role during cue and/or context exposure depends on a number of different factors, including the extent of cocaine training, brain region, and withdrawal duration (*Lu et al., 2005*; *Koya et al., 2009*; *Edwards et al., 2011*; *Whitfield et al., 2011*; *Sun et al., 2013*; *Doyle et al., 2014*). One finding of particular interest showed that following cocaine self-administration, bilateral infusion of U0126 into the basolateral amygdala immediately after a short memory retrieval session, attenuated subsequent cocaine seeking. This treatment was associated with a transient decrease in phospho-ERK, suggesting an important role for ERK during memory retrieval in maintaining drug-associated memories (*Wells et al., 2013*).

To date, the knowledge implicating ERK signalling in drug addiction has not been translated into clinically relevant applications due to the lack of drugs with low toxicity and an ability to efficiently cross the blood-brain barrier. Here, we report on two new cell-penetrating peptides able to attenuate the activation of this signalling cascade in response to cocaine. We further demonstrate that these two peptides prevent the formation of drug-associated memories in the CPP. We also demonstrate that PD325901, a potent MEK inhibitor already in clinical trials for cancer, was able to penetrate the brain and efficiently block Ras-ERK activation. Furthermore, a single in vivo administration of PD325901 persistently blocked a CPP response and significantly accelerated the extinction of cue-induced responding following cocaine self-administration.

## Results

### Two novel cell-penetrating peptides inhibit the ERK signalling pathway

Cell-penetrating peptides have been shown to hold good promise for the treatment of neuropsychiatric disorders, in particular for their generally low toxicity and tolerability (*Ramsey and Flynn, 2015*; *Raucher and Ryu, 2015*). Despite their potency is typically in the micromolar range, their ability to partially disrupt protein–protein interactions instead of blocking the enzymatic activity is also a potential advantage. In addition, cell-penetrating peptides dissolve easily in water solutions and do not require organic solvents, a major complication in the formulation of many small therapeutic molecules, including the more potent PD325901.

Bearing in mind these observations, we have designed and generated two active cell-penetrating peptides, based on protein-protein interaction sequences found in Ras-ERK components, able to attenuate the activation of this signalling cascade in vivo. The first peptide, hereafter referred to as RB1 (*Figure 1A*), was designed around the 59–73 portion, the KIM sequence, of the ERK1/2 specific phosphatase MKP3 (*Liu et al., 2006*). The KIM sequence is responsible for the interaction with the common docking (CD) domain of ERK1/2. The second peptide, hereafter referred to as RB3 (*Figure 1B*), was designed with the support of molecular graphics tools, taking advantage of the published crystal structure of a ternary Ras:SOS:Ras*GDP complex (*Sondermann et al., 2004*),

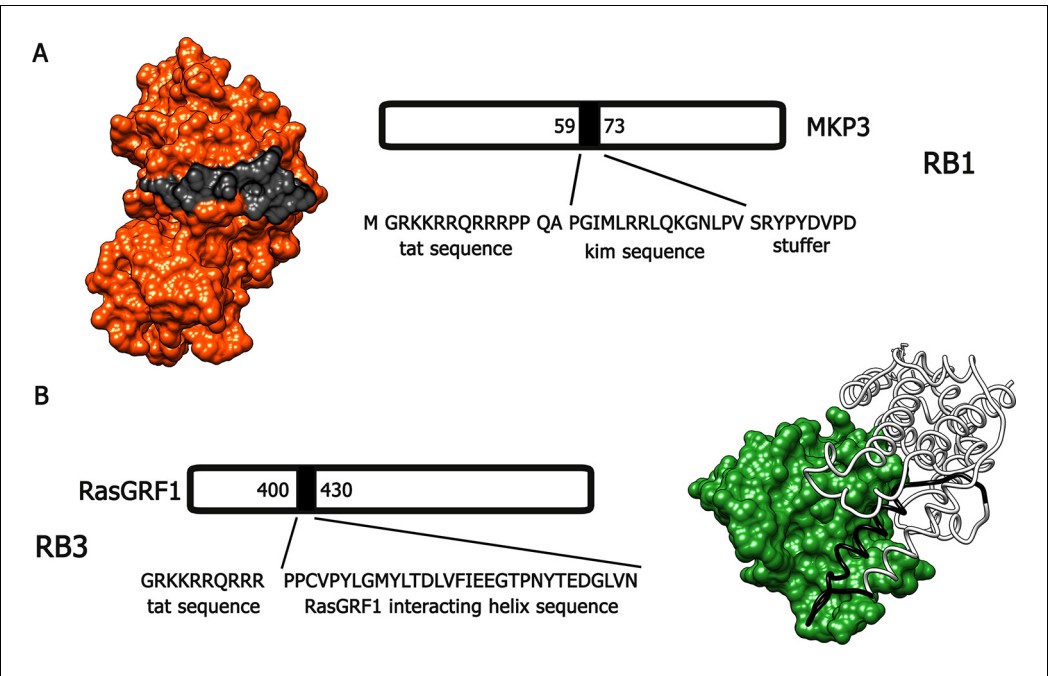

**Figure 1.** Molecular modeling of RB1 and RB3. (**A**) RB1 peptide (in black) is shown bound to ERK (in red). Graphic was made using UCSF Chimera software, PDB id. 2FYS. (**B**) RB3 peptide (black ribbon) is shown bound to Ras (green). Graphic was made using UCSF Chimera software, aligning the crystal structure of the CDC25 domain of RasGRF1 (PDB id. 2IJE) with the crystal structure of a ternary Ras:SOS:Ras*GDP complex (PDB id.1XD2).

compared to the crystal structure of the CDC25 domain of Ras-GRF1, a neuronal specific Ras guanine nucleotide exchange factor (Ras-GEF). The peptide sequence (1173–1203 portion of the CDC25 domain) includes an α-helix crucial for the GTP exchange activity on RAS proteins. To allow these peptides to cross cellular membranes and the blood-brain barrier, we added to their sequences a portion of the HIV TAT protein (*Gump and Dowdy, 2007*), thus generating two distinct cell-penetrating peptides (*Papale et al., 2012*).

We first tested these peptides for their inhibitory potentials on ERK phosphorylation after stimulation with 100 µM glutamate in an ex-vivo model of acute striatal brain slices. As a control, we used a scrambled, inactive, peptide sequence (Scrambled, Scr). RB1 significantly reduced ERK phosphorylation starting from 20 µM with a half-maximal inhibitory concentration (IC50) of 17.25 µM (*Figure 2A*), while RB3 significantly reduced ERK activity starting from 10 µM with an IC50 of 6 µM (*Figure 2B*).

We then investigated whether RB1 and RB3 may also affect the phosphorylation of (Ser10)-acetylated (Lys14) histone H3 (pAc-H3) and S6 ribosomal protein, (pS6, Ser235/236 specific site), two well-characterised ERK substrates (*Santini et al., 2007*, *2009*; *Darmopil et al., 2009*; *Orellana et al., 2012*). RB1 was effective in reducing pAc-H3 with an IC50 of 1.12 µM (*Figure 2C*). Similarly, RB3 decreased the phosphor-ylation of Ac-H3 with an IC50 of 5.2 µM (*Figure 2D*). Both peptides were also able to reduce pS6 levels with an IC50 of 4.02 µM for RB1 (*Figure 2E*) and 3.69 µM for RB3 (*Figure 2F*), respectively.

We then tested these two peptides in vivo by measuring their ability to block the phosphorylation of ERK and downstream substrates upon an acute cocaine administration. Mice were pre-treated with a single dose (20 mg/kg i.p.) of either peptide or scrambled control 1 hr prior to a cocaine injection (25 mg/kg). This peptide dose was selected based on previous work (*Borsello et al., 2003*; *Besnard et al., 2011*; *Papale et al., 2012*). Five minutes after cocaine administration, mice were

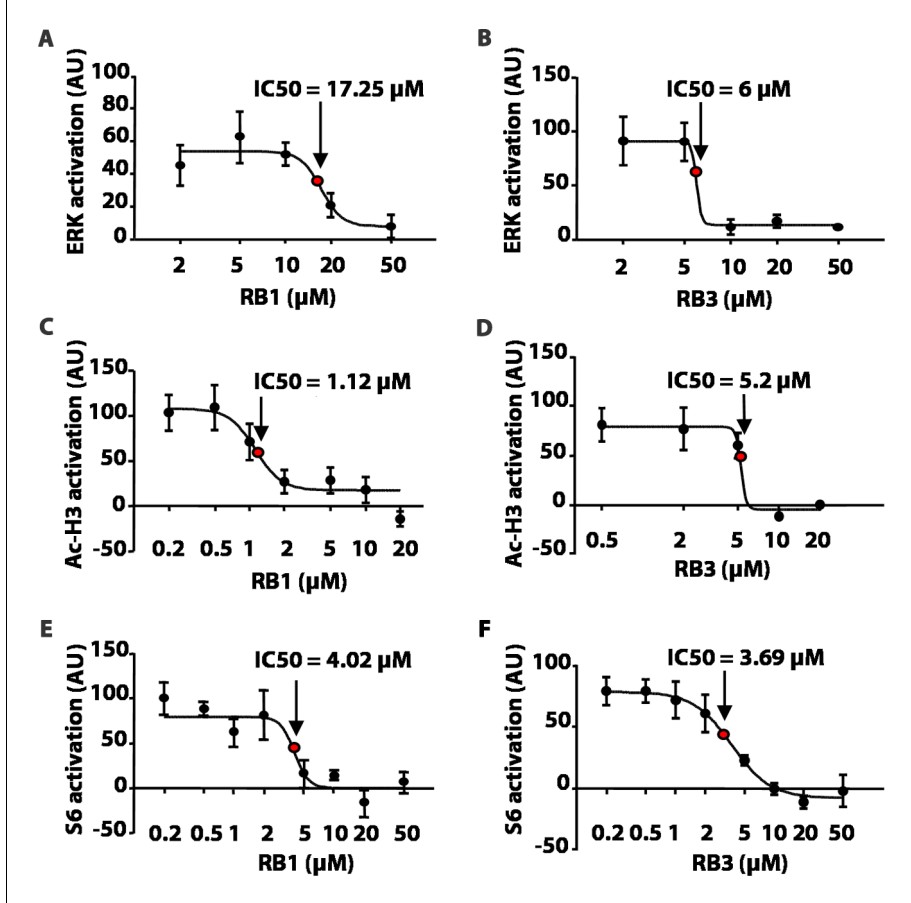

**Figure 2.** RB1 and RB3 inhibit ERK, histone H3 and S6 ribosomal protein activation in an ex-vivo model of acute striatal slices in a dose-dependent manner. (A–F) Dose response curves of RB1 and RB3 for ERK, Ac-H3 and S6 activation. 200 $\mu$m thick striatal slices were freshly prepared from 2-month old mice and transferred into a perfusion chamber for 1 hr at 32°C. Slices were pre-treated with either different doses of peptides (RB1 and RB3) or scrambled controls (scramble RB1 or scramble RB3). After 1 hr, one group of slices for each condition was fixed in PFA 4% for 15 min, while another group was stimulated with glutamate 100 $\mu$M for 10 min prior to fixation. 18 $\mu$m cryo-sections were processed for immunohistochemistry with anti-phospho p44/p42 MAP kinase (Thr202/Tyr204) or for immunofluorescence co-labelling the slices with anti-phospho (Ser10) - acetyl (Lys14) H3 or anti-phospho S6 (Ser235/236) and the neuronal marker NeuN. Neuronal quantification was performed with ImageJ software by counting the number of phospho-ERK positive cells and the number of phospho-Ac-H3 or phospho-S6 among NeuN positive neurons in each slice. The level of activation is expressed on the Y-axis as arbitrary units (AU). Doses are reported in a logarithmic scale (Log10) on the X-axis. The IC50 was calculated for each specific response using GraphPad Prism software.

transcardially perfused and brains were harvested. A significant decrease of ERK phosphorylation was observed in the ventral striatum of both RB1- and RB3- treated mice (*Figure 3*). In further experiments we examined the phosphorylation of S6 and Ac-H3 in RB1- and RB3-treated groups 20 min following cocaine administration. Phosphorylation of S6 was reduced in RB3- but not RB1-treated mice (*Figure 4*), whereas a complete prevention of phospho-Ac-H3 was observed in both RB1- and RB3-treated mice (*Figure 5*).

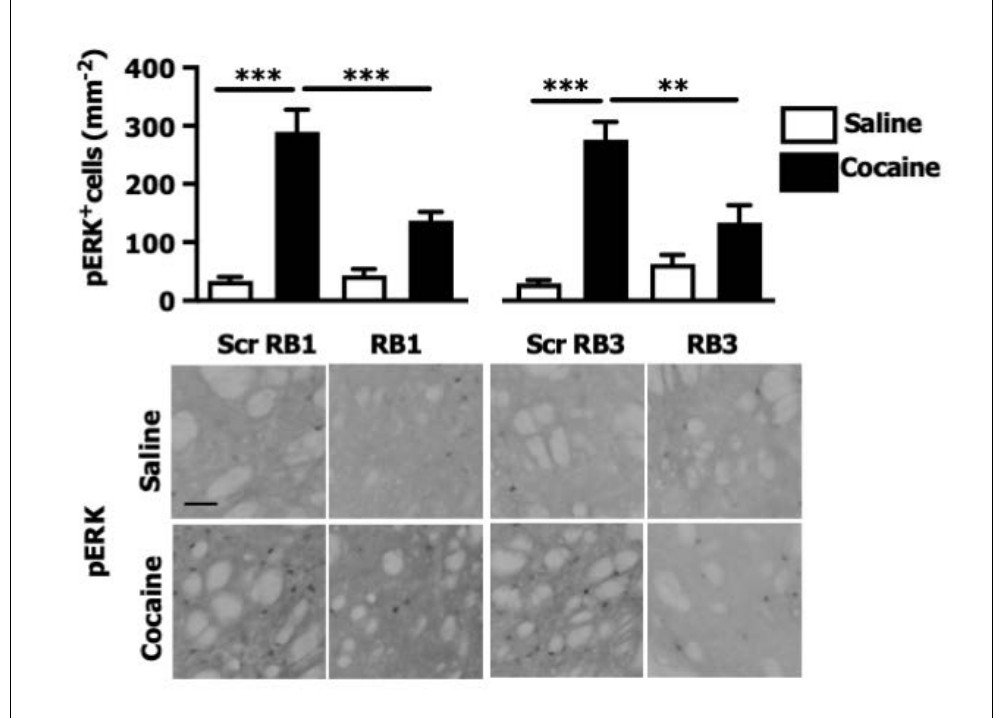

**Figure 3.** RB1 and RB3 have an inhibitory effect on cocaine-induced ERK phosphorylation. RB1 (20 mg/kg), RB3 (20 mg/kg), or the scrambled peptides (Scr) were administered to wild-type mice 1 hr before an acute cocaine (25 mg/kg) or saline injection. After 5 min mice were perfused. Immunohistochemistry was performed with anti-phospho p44/42 MAP kinase (Thr202/Tyr204, scale bar 30 $\mu$m). Quantification of phospho-ERK positive cells shows a significant inhibitory effect of RB1 and RB3 on cocaine-induced ERK activation in the ventral striatum. Two-way ANOVA: effect of RB1 $F_{1,14}$ = 18.73, p<0.001, effect of cocaine $F_{1,14}$ = 111.87, p<0.0001, effect of interaction $F_{1,14}$ = 23.95, p<0.001; Bonferroni's post-hoc, Scr RB1 saline (n=5) *vs* Scr RB1 cocaine (n=3) p<0.001, Scr RB1 cocaine (n=3) *vs* RB1 cocaine (n=5) p<0.001. Two-way ANOVA: effect of RB3 $F_{1,14}$ = 6.26, p<0.05, effect of cocaine $F_{1,14}$ = 52.11, p<0.0001, effect of interaction $F_{1,14}$ = 16.06, p<0.01; Bonferroni's post-hoc, Scr RB3 saline (n=5) *vs* Scr RB3 cocaine (n=3) p<0.001, Scr RB3 cocaine (n=3) *vs* RB3 cocaine (n=5) p<0.01. **p<0.01, ***p<0.001. Data are shown as mean with SEM.

## The MEK inhibitor PD325901 is able to effectively block the Ras-ERK pathway in the brain via systemic injection

In recent years, some MEK inhibitors and Raf/B-Raf inhibitors, able to block Ras-ERK signalling in vivo, have already been tested clinically for cancer therapy (*Uehling and Harris, 2015*; *Wu and Park, 2015*). Therefore, those drugs represent ideal candidates for repositioning studies to address efficacy also for neuropsychiatric disorders such as drug addiction. In order to verify whether these clinically relevant inhibitors pass the blood-brain barrier, we tested the MEK1/2 inhibitors PD325901, Trametinib (GSK1102212) and Selumetinib (AZD6244), and the Raf inhibitor Dabrafenib (GSK2118436). The doses of the inhibitors, as indicated below, were selected on the basis of their previously reported effects on tumour formation (*Hennig et al., 2010*; *Gilmartin et al., 2011*; *Hofmann et al., 2012*; *King et al., 2013*). For all compounds we followed the same procedure: the inhibitor was given (i.p.) 1 hr prior a cocaine injection (25 mg/kg), and 5 min after cocaine administration mice were transcardially perfused, brains were harvested, and ERK phosphorylation was subsequently determined. An acute administration of PD325901 (25 mg/kg) completely abolished ERK phosphorylation in the ventral striatum (*Figure 6A*). In contrast, Trametinib (5 mg/kg) and Selumetinib (50 mg/kg) were not effective (*Figure 6B–C*), while Dabrafenib (50 mg/kg) had a partial but not significant effect on ERK phosphorylation in the ventral striatum (*Figure 6D*).

As PD325901 was the only compound to fully prevent ERK phosphorylation in the brain in vivo, we further investigated its properties by measuring the IC50 toward ERK phosphorylation. In the ex-

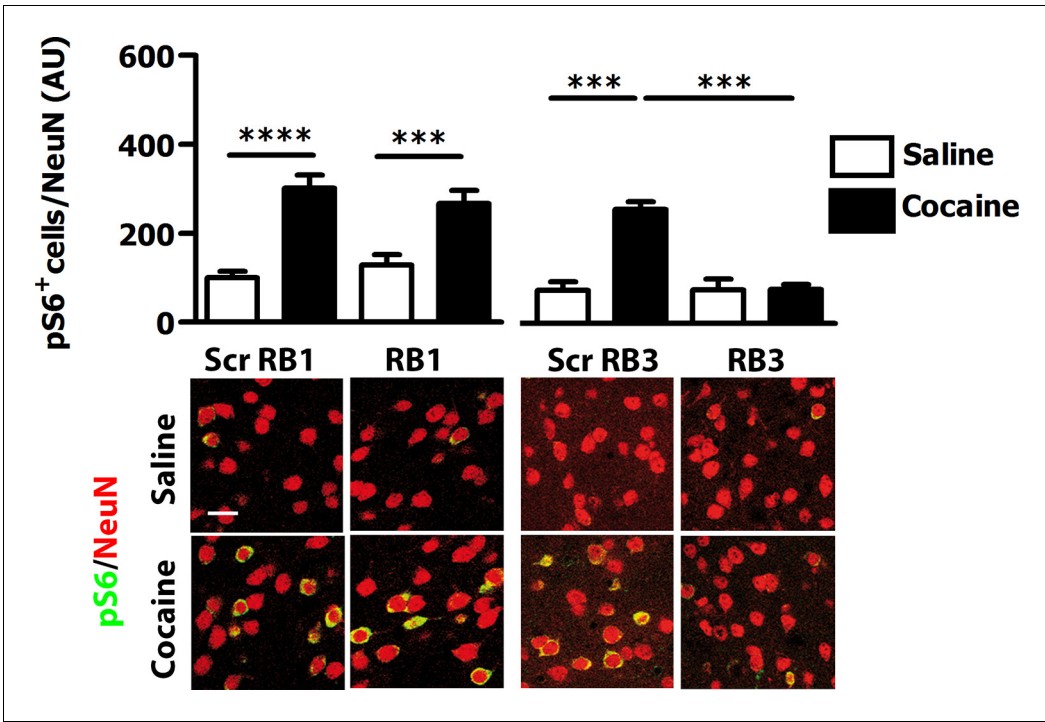

**Figure 4.** RB3, but not RB1 has an inhibitory effect on cocaine-induced S6 phosphorylation. RB1 (20 mg/kg, i.p.), RB3 (20 mg/kg, i.p.) or the scrambled peptides (Scr RB1 and Scr RB3, 20 mg/kg, i.p.) were administered to wild type mice 1 hr before an acute cocaine (25 mg/kg) or saline injection. After 20 min mice were perfused. Co-labelling was performed with anti-phospho S6 ribosomal protein (Ser235/236, in green) and NeuN (in red, scale bars 30 $\mu$m). Neuronal quantification shows that RB3 completely prevented cocaine-induced S6 phosphorylation in the ventral striatum. Two-way ANOVA: effect of RB3 $F_{1,14} = 25.29$, p<0.001, effect of cocaine $F_{1,14} = 26.30$, p<0.001, effect of interaction $F_{1,14} = 25.88$, p<0.001; Bonferroni's post-hoc, Scr RB3 saline (n=4) *vs* Scr RB3 cocaine (n=5) p<0.001, Scr RB3 cocaine (n=5) *vs* RB3 cocaine (n=5) p<0.001. RB1 does not exert any significant effect on cocaine-induced S6 activation. Two-way ANOVA: effect of RB1 $F_{1,36} = 0.01$, p>0.05, effect of cocaine $F_{1,36} = 46.63$, p<0.001, effect of interaction $F_{1,36} = 1.65$, p>0.5; Bonferroni's post-hoc, Scr RB1 saline (n=10) *vs* Scr RB1 cocaine (n=10) p<0.0001, Scr RB1 cocaine (n=10) *vs* RB1 cocaine (n=10) p<0.001. ***p<0.001, ****p<0.0001. Data are shown as mean with SEM.

vivo model of acute brain slices, already used for RB1 and RB3, we found that PD325901 showed an IC50 of 1.15 nM, similar to what previously reported by Pfizer (0.33 nM and 0.59 nM, for inhibiting MEK1 in a cell free system and blocking pERK in tumour cells, respectively) (*Brown et al., 2007*; *Barrett et al., 2008*) (*Figure 6—figure supplement 1*).

Next, we investigated the role of PD325901 in cocaine-mediated responses, without pursuing additional studies with the other compounds. We first determined in vivo the minimal effective dose necessary to abolish cocaine-induced ERK phosphorylation. We tested increasing concentrations of PD325901 (0.25 to 25 mg/kg), with complete blockade of ERK phosphorylation in vivo achieved at 2.5 mg/kg (*Figure 7*).

## Long-lasting impairment of conditioned place preference after a single administration of PD325901

The potent inhibition of ERK phosphorylation resulting from PD325901 prompted us to test its potential benefits in the treatment of cocaine-mediated processes in the CPP paradigm. In this classical Pavlovian conditioning procedure, the drug serves as an unconditioned stimulus (US) that is repeatedly paired with a distinct environment (conditioned stimulus, CS) within the testing chamber and the animal learns to associate this specific context with the rewarding effects of the drug (*Tzschentke, 2007*).

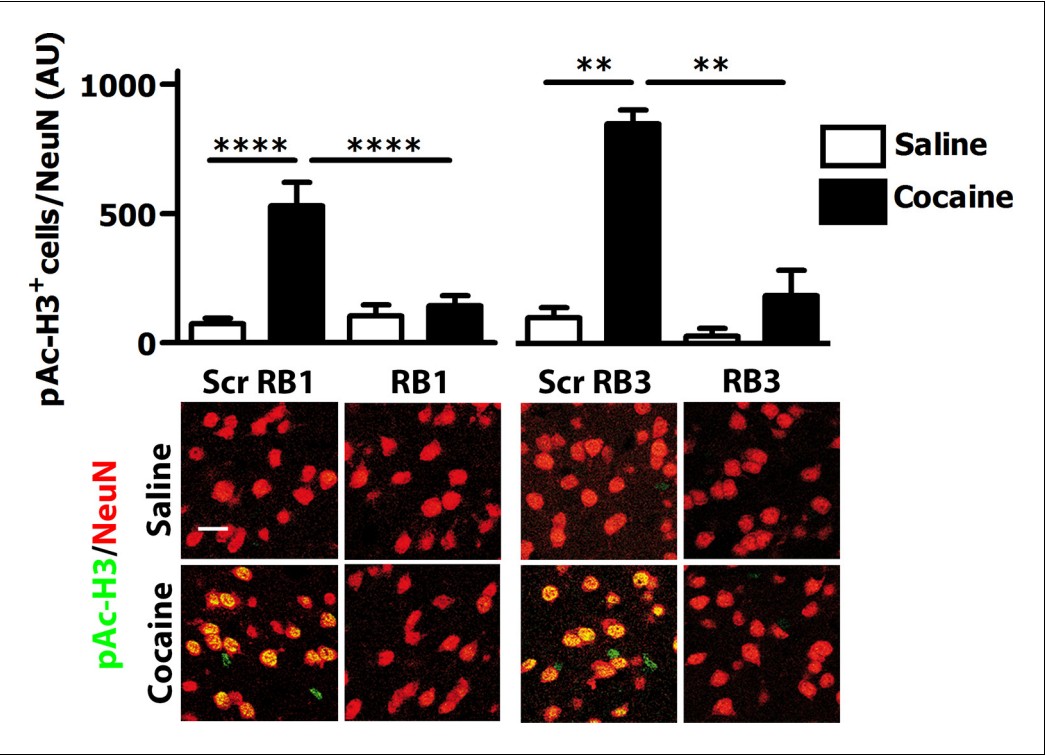

**Figure 5.** RB1 and RB3 have an inhibitory effect on cocaine-induced H3 phosphorylation. RB1 (20 mg/kg, i.p.), RB3 (20 mg/kg, i.p.) or the scrambled peptides (Scr RB1 and Scr RB3, 20 mg/kg, i.p.) were administered to wild type mice 1 hr before an acute cocaine (25 mg/kg) or saline injection. After 20 min mice were perfused. Co-labelling was performed with anti-phospho (Ser10)-acetyl (Lys14) histone H3 (in green) and NeuN (in red, scale bars 30 $\mu$m). Neuronal quantification shows a significant inhibitory effect of RB1 and RB3 on cocaine-induced H3 activation in the ventral striatum. Two-way ANOVA: effect of RB1 $F_{1,32}$ = 10.29, p<0.01, effect of cocaine $F_{1,32}$ = 19.91, p<0.001, effect of interaction $F_{1,32}$ = 13.99, p<0.001; Bonferroni's post-hoc, Scr RB1 saline (n=9) $vs$ Scr RB1 cocaine (n=9) p<0.0001, Scr RB1 cocaine (n=9) $vs$ RB1 cocaine (n=9) p<0.0001. Two-way ANOVA: effect of RB3 $F_{1,14}$ = 9.90, p<0.01, effect of cocaine $F_{1,14}$ = 14.84 p<0.01, effect of interaction $F_{1,14}$ = 6.09, p<0.05, Bonferroni's post-hoc, Scr RB3 saline (n=4) $vs$ Scr RB3 cocaine (n=5) p<0.01, Scr RB3 cocaine (n=5) $vs$ RB3 cocaine (n=5) p<0.01. **p<0.01, ****p<0.0001. Data are shown as mean with SEM.

We examined the effects of PD325901 on the retrieval of cocaine CPP memory (*Figure 8A*). Animals were trained for 6 days using an unbiased CPP protocol to associate one compartment with cocaine (20 mg/kg, i.p.) on day 2, 4, 6, while a different compartment was paired with saline on day 1, 3, 5. Control animals received saline in both compartments. Twenty-four hours following the last conditioning session, mice were given either a systemic injection of PD325901 (10 mg/kg, i.p.) or vehicle 1 hr before CPP testing (test 1, day 8). A significant preference for the cocaine-paired chamber was observed in mice pre-treated with vehicle, whereas PD325901-treated mice failed to demonstrate a conditioned response (*Figure 8B*).

To test the enduring effect of PD325901 on the cocaine memory, animals were retested in a drug-free state in the CPP apparatus 2 weeks later (day 22). While vehicle-treated animals continued to show a side preference, indicative of the persistence of the cocaine memory, PD325901-treated mice still lacked a conditioned cocaine preference (*Figure 8C*). On the following day (day 23), both groups of mice were re-exposed to a cocaine conditioning session in the same drug-paired environment to reactivate the drug-related memory acquired during previous training. 24 hr later, another test of preference was given (test 3, day 24). Cocaine re-exposure failed to affect PD325901-treated animals, while in vehicle-treated animals, a robust behavioural response was still observed (*Figure 8D*). We ruled out any possible confounding effect on locomotor functions as a consequence of PD pre-treatment by measuring the distance travelled in the CPP apparatus on day 1 and day 8.

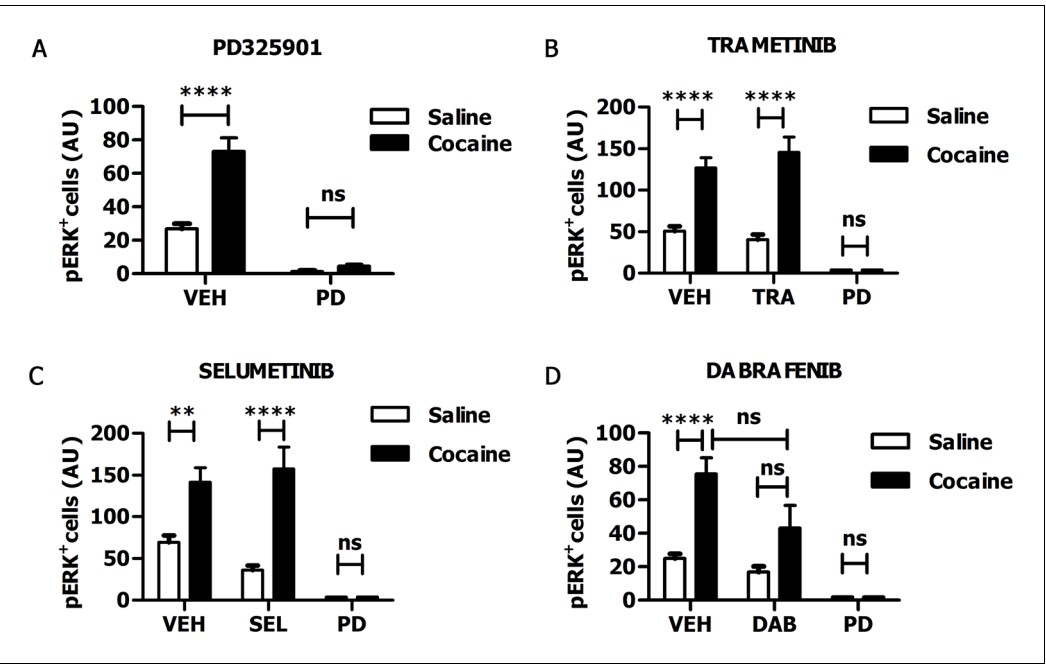

**Figure 6.** PD325901 prevents cocaine-induced ERK phosphorylation in vivo. Mice received an injection of different inhibitors or vehicle followed by cocaine (25 mg/kg, i.p.) or saline injection 1 hr later. 5 min after the stimulation, mice were perfused and ERK phosphorylation in the ventral striatum was determined. (**A**) PD325901 (25 mg/kg, i. p.) completely blocked ERK phosphorylation. Two-way ANOVA, effect of pre-treatment $F_{1,80}$ = 125.76 p<0.0001, effect of cocaine $F_{1,80}$ = 34.66 p<0.0001, effect of interaction $F_{1,80}$ = 26.25 p<0.0001; Bonferroni's post-hoc, PD Saline vs PD Cocaine: p>0.05, VEH Saline vs VEH Cocaine: p<0.0001. (**B**) Mice were pre-treated with Trametinib (GSK1102212) (5 mg/kg, i.p.), vehicle or PD325901 (25 mg/kg, i.p.) as a positive control for the inhibition. Trametinib did not prevent ERK phosphorylation. Two-way ANOVA, effect of pre-treatment $F_{2,69}$ = 39.06 p<0.0001, effect of cocaine $F_{1,69}$ = 49.26 p<0.0001, effect of interaction $F_{2,69}$ =11.51 p<0.001; Bonferroni's post-hoc, VEH Saline vs VEH Cocaine: p<0.0001; TRA Saline vs TRA Cocaine: p<0.0001; PD Saline vs PD Cocaine: p>0.05. (**C**) Mice were pre-treated with Selumetinib (AZD6266) (50 mg/kg), vehicle or PD325901 (25 mg/kg, i.p.) as a positive control for the inhibition. Selumetinib failed to prevent ERK phosphorylation. Two-way ANOVA, effect of pre-treatment $F_{2,72}$ = 20.91 p<0.0001, effect of cocaine $F_{1,72}$ = 24.49 p<0.0001, effect of interaction $F_{2,72}$ = 6.62 p<0.01; Bonferroni's post-hoc, VEH Saline vs VEH Cocaine: p<0.01; SEL Saline vs SEL Cocaine: p<0.0001; PD Saline vs PD Cocaine: p>0.05. (**D**) Mice were pre-treated with Dabrafenib (GSK2118436) (50 mg/kg), vehicle or PD325901 (25 mg/kg, i.p.) as a positive control for the inhibition. Dabrafenib displayed a partial, but not significant, inhibitory effect. Two-way ANOVA, effect of pre-treatment $F_{2,66}$ = 10.74 p<0.0001, effect of cocaine $F_{1,66}$ = 10.25 p<0.05, effect of interaction $F_{2,66}$ = 3.08 p>0.05; Bonferroni's post hoc, VEH Saline vs VEH Cocaine: p<0.0001; DAB Saline vs DAB Cocaine: p>0.05; VEH Cocaine vs DAB Cocaine: p>0.05. VEH=Vehicle, PD=PD325901, TRA=trametinib, SEL=Selumetinib, DAB=Dabrafenib, AU=arbitrary unit, ns=not significant, **p<0.01, ****p<0.0001. Data are shown as mean with SEM.

The following figure supplement is available for figure 6:

**Figure supplement 1.** PD325901 inhibits ERK phosphorylation in an ex-vivo model of acute striatal slices in a dose-dependent manner.

No difference among groups was observed (**Figure 8—figure supplement 1**). As additional confirmation, we monitored the horizontal activity of a different cohort of mice using locomotor activity cages equipped with an automatic infrared movement detector system. The activity was observed starting 30 min after an injection of PD325901 (10 mg/kg) and recorded every 5 min for a total of 6 trials. At the end of this session mice received a cocaine challenge (20 mg/kg, i.p.) and the effect of PD325901 pre-treatment on the hyper-locomotion induced by cocaine was evaluated for 30 min. No effects of PD325901 pretreatment were detected on spontaneous locomotion and cocaine-induced locomotion (**Figure 8—figure supplement 2**).

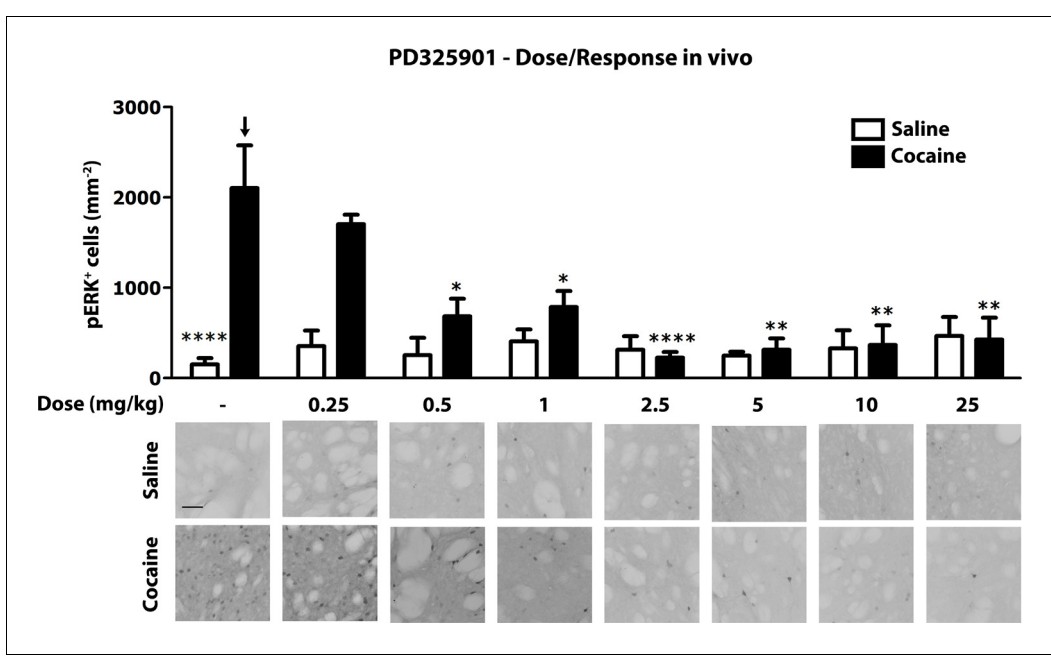

**Figure 7.** Dose response curve of MEK inhibitor PD325901 on ERK phosphorylation in vivo. Different doses of PD325901 were administered i.p. to wild-type mice 1 hr before the saline or cocaine (25 mg/kg) injection. After 5 min mice were perfused. Immunohistochemistry was performed with anti-phospho p44/42 MAP kinase (Thr202/Tyr204, lower panel, scale bar 30 $\mu$m). Quantification of phospho-ERK positive cells in the dorsal striatum (upper panel, mean with SEM) shows that PD325901 inhibited ERK in a dose-dependent manner. Two-way ANOVA, effect of PD325901 $F_{7,78} = 2.43$, $p<0.05$, effect of cocaine $F_{1,78} = 4.81$, $p<0.05$, effect of interaction $F_{7,78} = 3.47$, $p<0.01$; Bonferroni's post-hoc, vehicle saline (n=9) vs vehicle cocaine (n=15) $p<0.0001$, vehicle cocaine (n=15) vs PD325901 0.25 mg/kg (n=4) $p>0.05$, vehicle cocaine (n=15) vs PD325901 0.5 mg/kg (n=5) $p<0.05$, vehicle cocaine (n=15) vs PD325901 1 mg/kg (n=5) $p<0.05$, vehicle cocaine (n=15) vs PD325901 2.5 mg/kg (n=15), $p<0.0001$, vehicle cocaine (n=15) vs PD325901 5 mg/kg (n=5), $p<0.01$, vehicle cocaine (n=15) vs PD325901 10 mg/kg (n=5), $p<0.01$, vehicle cocaine (n=15) vs PD325901 25 mg/kg (n=5), $p<0.01$. *$p<0.05$, **$p<0.01$, ****$p<0.0001$, all the statistical significances are referred to the vehicle cocaine group indicated by the arrow. Data are shown as mean with SEM.

Furthermore, we investigated whether the inhibition of retrieval of cocaine-related memories in CPP was specifically due to MEK inhibition in the brain. Thus, we selected one compound from our previous experiment (see *Figure 6*), Trametinib, as a negative control for use in cocaine CPP. In contrast to PD325901, a single administration of 10 mg/kg of Trametinib did not interfere with the expression of CPP (*Figure 8—figure supplement 3*).

## Partial inhibition of RB1 and RB3 peptides on the retrieval of cocaine CPP

Despite the relevant effect of PD325901 on conditioned place preference, we decided also to test our RB1 and RB3 peptides in the same drug-related memory retrieval paradigm (*Figure 8—figure supplement 4A*). In this test we decided to use a mix of the two peptides, to obtain the highest inhibition possible of the Ras-ERK pathway. Mice received either a systemic injection of an RB1/RB3 mix (20 mg/kg, i.p.) or scrambled control 1 hr prior to CPP testing (test 1). As expected, control mice showed a robust preference for the cocaine-paired compartment, while mice pre-treated with the RB1/RB3 mix failed to demonstrate a cocaine preference (*Figure 8—figure supplement 4B*). Animals were retested 2 weeks later (test 2) in a drug-free state, during which preference for the cocaine-paired environment was significantly lower than that of control mice (*Figure 8—figure supplement 4C*). The day following test 2 (day 23), mice underwent a conditioning session in the CPP apparatus immediately after cocaine or saline injection. Cocaine CPP was tested the following day (day 24, test 3). RB1/RB3 pre-treated group showed only a partial preference for the cocaine side compared to vehicle group (*Figure 8—figure supplement 4D*).

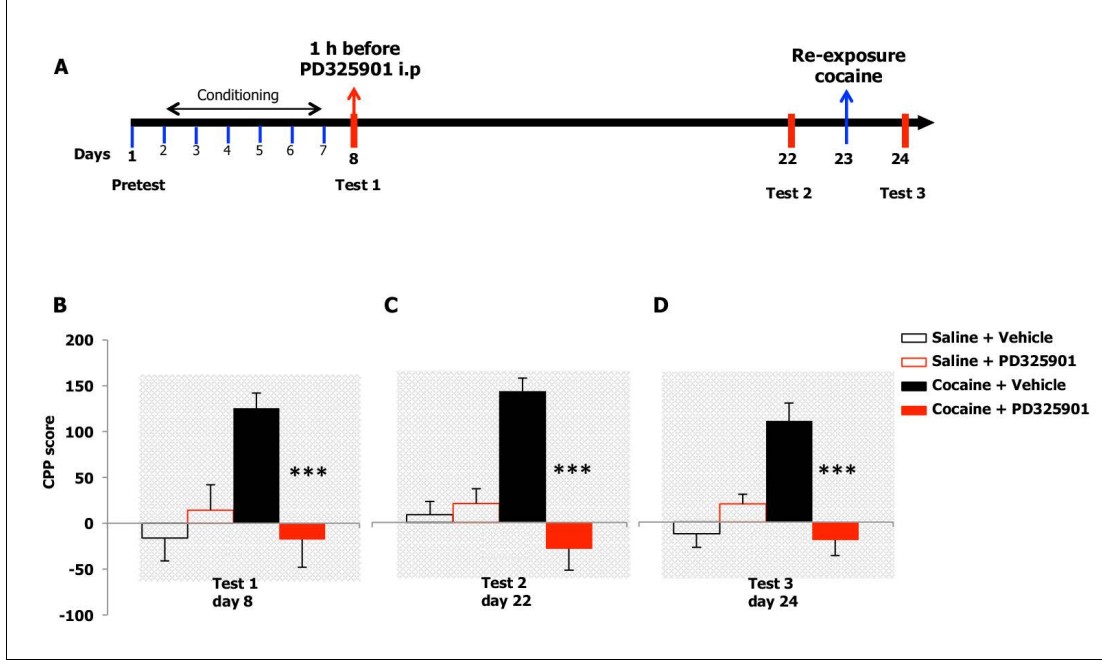

**Figure 8.** A single dose of PD325901 at 10 mg/kg blocks the retrieval of cocaine-associated memory in the conditioned place preference paradigm. (**A**) Experimental design of experiments reported in **B**, **C** and **D**. (**B**) The rewarding properties of cocaine (20 mg/kg i.p.) were first measured on day 8 (test 1). Mice treated with cocaine or saline during conditioning were further divided in 2 groups and received an acute injection of the MEK inhibitor, PD325901 (10 mg/kg i.p.), or DMSO vehicle 1 hr before test 1. A single administration of PD325901 fully inhibited CPP. A two-way ANOVA revealed a significant effect of drug $F_{1,35} = 4.305$, $p<0.05$, effect of PD325901 $F_{1,35} = 4.534$, $p<0.05$ and the interaction between these two factors $F_{1,35} = 10.713$, $p<0.001$; Bonferroni's post-hoc, cocaine PD325901 (n=12) vs cocaine vehicle (n=12), $p<0.0001$; cocaine vehicle (n=12) vs saline vehicle (n=8), $p<0.01$. (**C**) Two weeks later (test 2) mice were re-tested in the place conditioning apparatus in a drug-free state. In mice that were previously administered PD325901, a lack of preference is still observed. A two-way ANOVA revealed a significant effect of drug $F_{1,35} = 4.748$, $p<0.05$, effect of PD325901 pre-treatment $F_{1,35} = 16.877$, $p<0.0001$ and interaction between these two factors $F_{1,35} = 22.248$, $p<0.0001$; Bonferroni's post-hoc, cocaine PD325901 (n=12) vs cocaine vehicle (n=12), $p<0.0001$; cocaine vehicle (n=12) vs saline vehicle (n=8), $p<0.0001$; cocaine vehicle (n=12) vs saline PD (n=7), $p<0.001$. (**D**) One day 23, mice underwent a cocaine (20 mg/kg, i.p.) re-exposure conditioning session in the drug-paired compartment. 24 hr later (day 24, test 3), animals were retested for preference in a drug- and PD325901-free state. Re-exposure to cocaine in the drug-paired compartment was not able to rescue the expression of cocaine preference in PD325901-treated mice. A two-way ANOVA revealed a significant effect of drug $F_{1,35} = 5.100$, $p<0.05$, effect of PD325901 pre-treatment $F_{1,35} = 7.069$, $p<0.01$, and the interaction between these two factors $F_{1,35} = 19.404$, $p<0.0001$; Bonferroni's post-hoc, cocaine PD325901 (n=12) vs cocaine vehicle (n=12), $p<0.0001$; cocaine vehicle (n=12) vs saline vehicle (n=8), $p<0.0001$. ***$p<0.0001$. Data are shown as mean with SEM.

The following figure supplements are available for figure 8:

**Figure supplement 1.** PD325901 does not influence levels of locomotion during the conditioned place preference.

**Figure supplement 2.** Acute administration of PD325901 at 10 mg/kg had no effect on locomotion.

**Figure supplement 3.** TRAMETINIB does not interfere with the retrieval of cocaine-associated memory in the conditioned place preference paradigm.

*Figure 8 continued on next page*

*Figure 8 continued*

**Figure supplement 4.** Retrieval of cocaine place preference is partially inhibited by RB1 and RB3 peptides.

## PD325901 attenuated cue responding following cocaine self-administration

To examine the potential therapeutic effect of PD325901 in a more clinically relevant paradigm, we also attempted to determine whether PD325901 would produce a reduction in cocaine seeking when administered prior to a short cue-only retrieval session following cocaine self-administration in mice. Mice underwent daily 1-hr cocaine self-administration for 7 consecutive days (days 1–7) with no prior lever training. Cocaine (0.50 mg/kg in a 14 µl infusion) delivery was contingent upon pressing on the active lever under an FR1 schedule of reinforcement and paired with the 20 s presentation of a blinking light stimulus (CS), which also served as a timeout period, during which lever presses were not reinforced. Presses on the inactive lever were recorded but had no scheduled consequence. Twenty-four hours following the last cocaine self-administration session, mice were given a single 10-minutes CS-only retrieval session (day 8), preceded 30 min by a vehicle (10 ml/kg) or PD325901 (10 mg/kg, i.p.) injection, during which presses on the active lever were not reinforced, but the CS was presented. Beginning the following day, mice were presented with 10 daily 1-hr extinction sessions (days 9–18), during which presses on the active lever were not reinforced, but the CS was presented, to determine the effect of PD325901 administered during the short retrieval trial on subsequent cue responding. *Figure 9A* shows the timeline of the self-administration protocol. Prior to vehicle or MEK inhibitor treatment on day 8, both groups demonstrated similar acquisition of cocaine self-administration (days 1–7), as demonstrated by a similar number of lever presses (*Figure 9B*) and number of cocaine reinforcers achieved (*Figure 9C*) during 7 daily 1-hr sessions of cocaine SA (0.50 mg/kg/infusion). Furthermore, responding for the CS in the absence of cocaine during the 10-min retrieval trial preceded by vehicle or MEK inhibitor administration on day 8 also did not differ. However, mice pre-treated with PD325901 demonstrated a significant decrease in active lever responding relative to vehicle controls across 10 daily 1-hr CS-only extinction sessions (days 9–18; *Figure 9D*). These data indicate that PD325901 administered prior to a brief CS-only retrieval trial had long-lasting consequences and attenuated subsequent responding in the presence of the CS.

## 10 mg/kg of PD325901 does not cause memory impairments in novel object recognition and inhibitory avoidance tests

Maladaptive cognitive mechanisms are at the basis of addiction. In particular, it is well established that CPP relies on Pavlovian conditioning while self-administration is a form of instrumental learning. Indeed, inhibition of the Ras-ERK signalling pathway results in long-term memory deficits in a number of behavioural tests in rodents (*Adams and Sweatt, 2002*; *Davis and Laroche, 2006*; *Fasano and Brambilla, 2011*).

In order to verify whether 10 mg/kg of PD325901 may interfere with memory processes in general, we performed two well-established memory tasks, the novel object recognition (NOR) and the inhibitory avoidance (IA), which previous work has shown to be sensitive to MEK inhibition (*Kelly et al., 2003*; *Goeldner et al., 2008*; *Cestari et al., 2014*). In these tasks, we assessed the effect of PD325901 on both memory consolidation and retrieval.

Firstly, we tested memory acquisition in the NOR paradigm: after a short habituation session in the empty arena on day 1, mice were pre-treated with 10 mg/kg of PD325901 one hour before the training session on day 2, and then tested for their 24 hr long-term memory (*Figure 10A*). PD325901 had no effect on the basal preference or the motivational state of the animals (*Figure 10B–C*), and the percentage of time exploring the novel object was significantly higher than the percentage of time exploring the familiar object in both groups (*Figure 10D*). The discrimination index (DI) of the PD325901 treated group was equivalent to that of the vehicle treated group (*Figure 10E*). Furthermore, we calculated also the recognition index (RI), a recently described index that takes into account possible biases due to spatial preferences (*d'Isa et al., 2014*). While the DI measures the ability to discriminate between two objects presented at a same time point, the RI

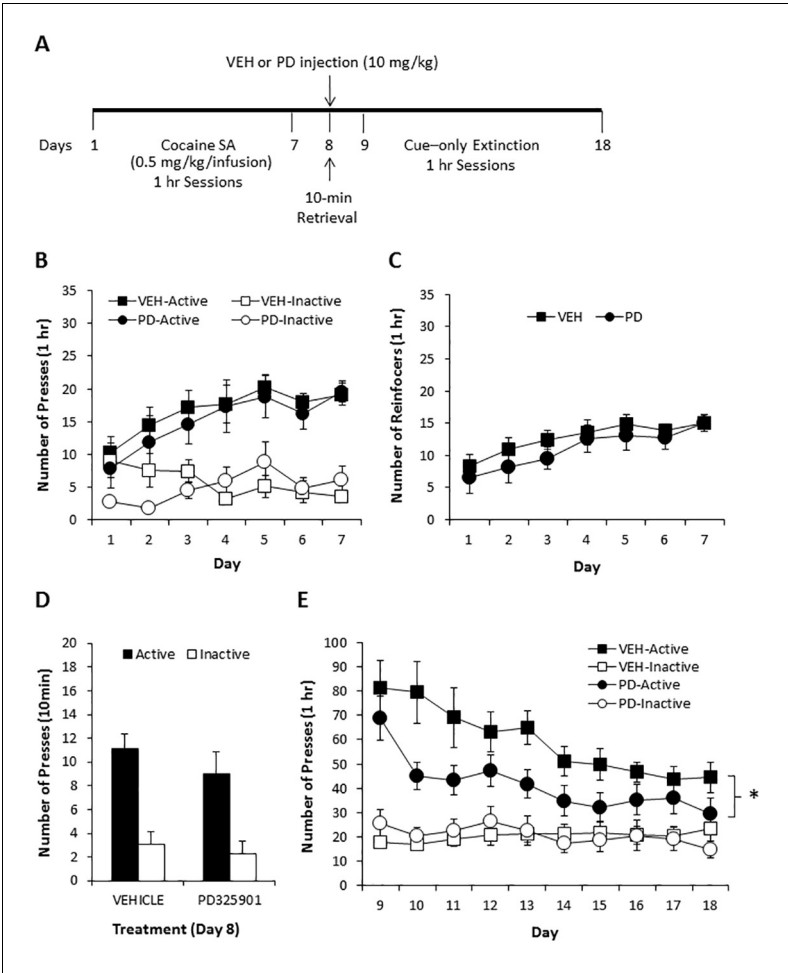

**Figure 9.** A single dose of PD325901 at 10 mg/kg attenuates cue responding following cocaine self-administration. (**A**) Experimental design of cocaine self-administration experiments reported in **B**, **C**, **D**, and **E**. (**B**) Mice first underwent 7d of daily 1 hr cocaine self-administration sessions (days 1–7). There were no differences in cocaine self-administration based on treatment group prior to drug manipulations, as indicated by a similar number of lever presses [a three-way ANOVA (lever x day x treatment) revealed significant main effects of lever $F_{1,29} = 115.048$, p<0.0005 and day $F_{3.5,101.1} = 3.460$, p<0.05, and a significant lever x day interaction $F_{3.7,106.9} = 4.080$, p<0.05, but no other significant effects (p>0.05)] and (**C**) reinforcers [a two-way ANOVA (day x treatment) revealed a significant effect of day $F_{3.1,89.4} = 6.341$, p<0.005, but no other effects (p>0.05)]. (**D**) DMSO vehicle (10 ml/kg i.p., n = 18) or PD325901 (10 mg/kg i.p., n = 13) was administered 30 min prior to a 10-min retrieval trial on day 8, during which lever pressing resulted in the CS but no cocaine. No difference in the groups was observed. A two-way ANOVA (lever x genotype) revealed a main effect of lever $F_{1,29} = 48.429$, p<0.0005, but no other significant effects (p>0.05). (**E**) During 10 subsequent daily 1 hr CS-only sessions, PD325901-treated mice demonstrated decreased responding for the cocaine-associated CS. A three-way ANOVA (lever x day x treatment) revealed a significant lever x treatment interaction $F_{1,29} = 5.090$, p<0.05, with follow-up ANOVAs revealing a significant difference between responding on the active $F_{1,29} = 4.250$, p<0.05, but not inactive $F_{1,29} = 0.007$, p=0.932, lever. *p<0.05. Data are shown as mean with SEM.

quantifies the ability to recognize an object across different time points. The RI of the PD325901 treated group was equivalent to that of the vehicle treated group (*Figure 10F*). We next repeated the same experiment using a higher dose of PD325901 (25 mg/kg). Neither basal preference nor motivational state of the animals was affected (*Figure 10G–H*). In contrast, for the PD325901 treated-group percentages of time spent exploring the familiar object were closer to chance level (50%) than in controls (*Figure 10I*). As a consequence, both DI and RI were found significantly lower in PD325901 treated animals than vehicle treated animals (*Figure 10J–K*). These data indicate that

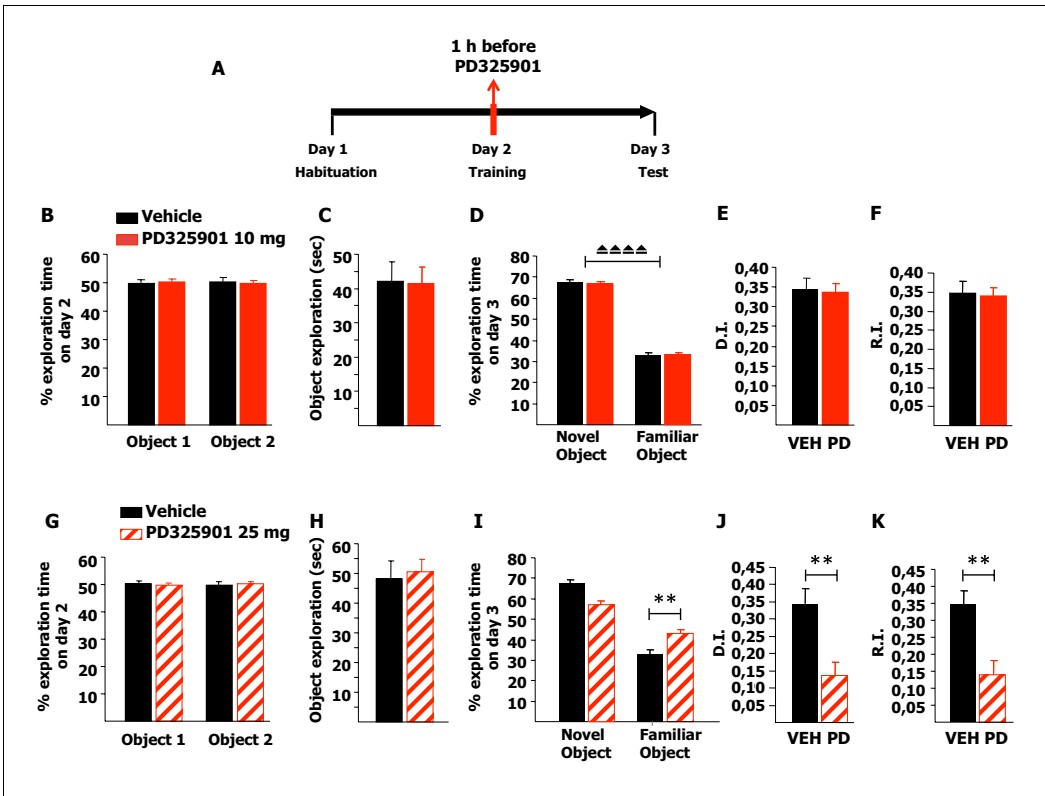

**Figure 10.** A single dose of PD325901 at 10 mg/kg does not affect acquisition of the Novel Object Recognition. (**A**) After a 5 min habituation session in the empty arena on day 1, mice were injected with PD325901 (10 or 25 mg/kg) or vehicle one hour before the 10 min training session with two identical objects on day 2 and were then evaluated for their 24 hr long-term memory in a 10 min testing session on day 3. (**B**) PD325901 10 mg/kg had no effect on basal preferences, as for both groups percentages of exploration of the two objects resulted to be comparable (Paired-samples t-test, PD325901: t9 = 0.220, p=0.831; VEH: t9 = −0226, p=0.827). (**C**) Total time of object exploration was equivalent between groups, meaning that PD325901 injection had no influence on the motivational state of the animals (Independent-samples t-test: t18 = −0.083, p=0.935). (**D**) On day 3, animals underwent the test for their 24 hr long-term memory. For both groups, the percentage of time spent exploring the novel object was significantly higher than the percentage of time spent exploring the familiar object (Paired-samples t-test, PD325901: t9 = 14.788, p<0.0001; VEH: t9 = 13.623, p<0.0001), indicating a significant learning. (**E**) The discrimination index (D.I.) of the PD325901-injected group was similar to the vehicle injected group (PD325901: 0.337 ± 0.023, VEH: 0.346 ± 0.025; Independent-samples t-test: t18 = −0.258, p=0.799) and above chance level (One-sample t-test, PD325901: t9 = 14.819, p<0.0001; VEH: t9 = 13.615, p<0.0001), indicating that 10 mg/kg of PD325901 did not affect the acquisition process in this memory test. (**F**) The recognition index (R.I.), based on the ability of an animal to recognize a same object at different time points, showed that the memory of the PD325901-treated group was equivalent to the vehicle treated group (PD325901: 0.339 ± 0.023, VEH: 0.347 ± 0.030; Independent-samples t-test: t18 = −2.17, p=0.830) and significantly above chance level (One-sample t-test, PD325901: t9 = 14.697, p<0.0001; VEH: t9 = 11.628, p<0.0001). PD325901 (n = 10), VEH (n =10). (**G**) PD325901 25 mg/kg had no effect on basal preferences, as for both groups percentages of exploration of the two objects were similar (Paired-samples t-test, PD325901: t9 = −0.250, p=0.808; VEH: t8 = 0.137, p=0.895). (**H**) Total time of object exploration was equivalent between groups, meaning that a higher dose of PD325901 had no impact on the motivational state of animals (Independent-samples t-test: t17 = 0.326, p=0.749). (**I**) On day 3, animals underwent the test for their 24 hr long-term memory. For both groups, the percentage of time spent exploring the novel object was higher than the percentage of time spent exploring the familiar object (Paired-samples t-test, PD325901: t9 = 3.551, p<0.01; VEH: t8 = 7.819, p<0.0001). Importantly, the time spent exploring the familiar object was significantly lower in the vehicle treated group than in the PD325901 treated group (Independent-samples t-test: t17 = 3.561, p<0.01) indicating that PD325901 injection partially perturbed the recognition of the familiar object. (**J**) The D.I. of the PD325901-injected group was lower, by 39.75%, indicating that 25 mg/kg of PD325901 significantly interfered with memory acquisition (PD325901: 0.137 ± 0.039, VEH: 0.345 ± 0.044; Independent-samples t-test: t17 = −3.560, p<0.01) but still above chance level (One-sample t-test, PD325901: t9 = 3.548, p=0.006; VEH: t8 = 7.818, p<0.0001). (**K**) Analysis of the R.I. showed that for both groups the percentage of time

*Figure 10 continued on next page*

*Figure 10 continued*

spent exploring the familiar object significantly decreased compared with the one observed during the first exposition to the same object (Paired-samples t-test, PD325901: t9 = 3.339, p=0.009; VEH: t8 = 9.798, p<0.0001), but the level of recognition memory in the PD325901-treated group was strongly diminished, by 39.77% (PD325901: 0.139 ± 0.042, VEH: 0.349 ± 0.037; Independent-samples t-test: t17 = −3.690, p<0.01), although still above chance level (One-sample t-test, PD325901: t9 = 3.287, p<0.01; VEH: t8 = 9.356, p<0.0001). PD325901 (n = 10), VEH (n = 9). ▲▲▲▲ p<0.0001, **p<0.01. Data are shown as mean with SEM.

while 10 mg/kg of PD325901 had no effect on memory consolidation at 24 hr, 25 mg/kg did result in memory consolidation impairment.

Next, we investigated whether 10 mg/kg of PD325901 influenced the retrieval of recognition memory. In this experimental design, mice explored two identical objects on day 2 and received an injection of PD325901 (10 mg/kg) one hour before the test session with two different objects on day 3 (*Figure 11A*). On day 2, no statistically significant difference was present between the basal preferences (*Figure 11B*). On day 3, PD325901 did not alter the exploratory behaviour and both groups spent more time exploring the novel object, indicating a significant learning (*Figure 11C–D*).

The DI and RI of the PD325901-injected group were equivalent to the control group, suggesting that the retrieval of recognition memory was intact (*Figure 11E–F*). Using a higher dose of PD325901 (25 mg/kg) no basal preferences were evident in both groups (*Figure 11G*). On day 3, one hour after PD325901 injection, both groups explored the novel object to a greater extent, although in the PD325901 group recognition of the familiar object was partially perturbed (*Figure 11H*), despite the motivational state of the animals being normal (*Figure 11I*). The DI and RI of the PD325901-injected group was lower, by more than 20% indicating that this dose of PD325901 significantly interfered with the retrieval of recognition memory (*Figure 11J–K*).

We next explored the effects of the two different doses of PD325901 (10 or 25 mg/kg) on the consolidation and retrieval of fear memory using the Inhibitory avoidance test. Animals were placed in a lighted arena and when they entered the dark compartment, they received an aversive foot shock. Immediately after training, animals received an injection of PD325901 (10 or 25 mg/kg, i.p.) and were then evaluated for their 24 hr memory on the subsequent day (*Figure 12A*). 10 mg/kg of PD325901 did not perturb the consolidation of the inhibitory-avoidance learning, since at this dose both groups showed a significant increase in latency, whereas 25 mg of PD325901 significantly disrupted memory consolidation (*Figure 12B*). This dose-dependent effect was also observed when the compound was injected one hour before the test session, to evaluate its effect on memory retrieval (*Figure 12C*). Animal injected with a dose of 10 mg/kg of PD325901 showed a significant increase in latencies, while the group treated with a dose of 25 mg/kg did not (*Figure 12D*).

## Discussion

Cocaine use remains a global public health concern and is associated with significant medical and psychiatric comorbidities. Although knowledge of reward mechanisms has significantly increased in recent decades, there are currently no approved medications for the treatment of cocaine dependence and behavioural treatment has shown limited efficacy (*Shorter et al., 2015*).

Among the potential target candidates, the Ras-ERK signalling pathway holds a unique position. First, it is a well-characterised cascade, highly studied in the context of cancer therapy and in neuroscience. Second, it is selectively activated by cocaine in D1 expressing medium spiny neurons of the striatum, the same cells implicated in the behavioural responses to this drug. Third, its inhibition in the brain leads to blockade of cocaine-mediated responses. Fourth, its implication in both memory consolidation and reconsolidation processes predicts a relevant adjuvant role in the definition of behavioural therapies to treat addiction (*Fasano and Brambilla, 2011*; *Cerovic et al., 2013*; *Pascoli et al., 2014*; *Garcia-Pardo et al., 2016*).

To date, the major limitation to a therapeutic approach to treat cocaine dependence based on Ras-ERK inhibition has been related to the lack of suitable, safe drugs, able to efficiently pass the blood-brain barrier. Here, we show an initial characterization of three compounds, two newly designed cell permeable peptides and a small molecule, able to reach the brain, efficiently block

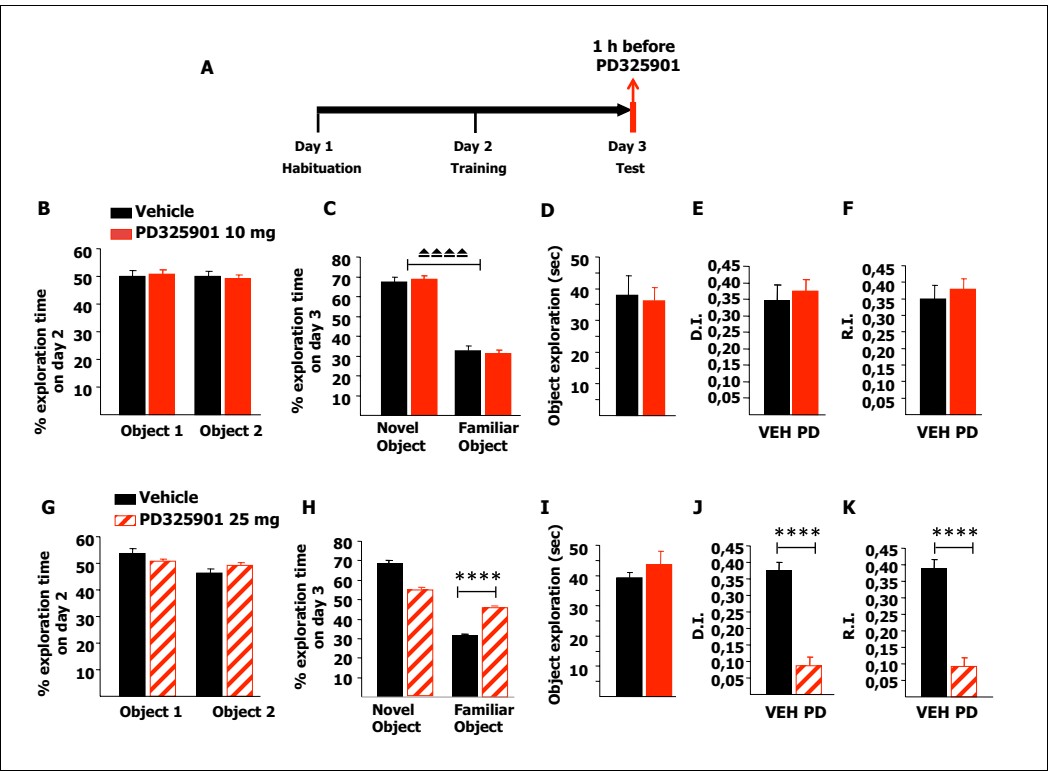

**Figure 11.** A single dose of PD325901 at 10 mg/kg does not affect retrieval of the Novel Object Recognition. (A) After a 5 min habituation session in the empty arena on day 1, mice were allowed to explore two identical objects in a 10 min training session on day 2. On the subsequent day, one hour before the test session mice received an injection of PD325901 (10 or 25 mg/kg) or vehicle and were then evaluated for their 24 hr long-term memory. (B) On day 2, no statistically significant difference was present between the basal preferences, as for both groups percentages of exploration of the two objects resulted to be comparable (Paired-samples t-test, PD325901: t9 = 0.587, p=0.572; VEH: t8 = 0.059, p=0.955). (C) On day 3, PD325901 did not alter the exploratory behaviour as the percentage of time spent exploring the novel object was significantly higher than the percentage of time spent on the familiar object for both groups, (Paired-samples t-test, PD325901: t9 = 10.886, p<0.0001; VEH: t8 = 7.424, p<0.0001), indicating a significant learning. (D) Total time of object exploration was equivalent between groups, meaning that PD325901 injection had no influence on the motivational state of the animals (Independent-samples t-test: t17 = −0.264, p=0.795). (E) The discrimination index (D.I.) of the PD325901-injected group was similar to the Vehicle injected group (PD325901: 0.375 ± 0.034, VEH: 0.347 ± 0.047; Independent-samples t-test: t17 = 0.493, p=0.628) and above chance level for both groups (One-sample t-test, PD325901: t9 = 10.871, p<0.0001; VEH: t8 = 7.426, p<0.0001), indicating that the retrieval of recognition memory was intact. (F) The recognition index (R.I.) showed that the recognition memory of the PD325901-treated group was equivalent to Vehicle treated group (PD325901: 0.380 ± 0.030, VEH: 0.349 ± 0.041; Independent-samples t-test: t17 = 0.617, p=0.545) and significantly above chance level (One-sample t-test, PD325901: t9 = 12.667, p<0.0001; VEH: t8 = 8.466, p<0.0001). PD325901 (n = 10), VEH (n = 9). (G) On day 2, no statistically significant difference was present between the basal preferences, as for both groups percentages of exploration of the two objects resulted to be comparable (Paired-samples t-test, PD325901: t9 = 0.813, p=0.437; VEH: t8 = 2.176, p=0.061). (H) On day 3, animals received an injection of 25 mg/kg of PD325901 and one hour later were evaluated in a 10 min session. For both groups, the percentage of time spent exploring the novel object was significantly higher than the percentage of time spent on the familiar object (Paired-samples t-test, PD325901: t9 = 3.286, p<0.01; VEH: t8 = 15.233, p<0.0001). Nevertheless, the time spent exploring the familiar object was significantly lower in the vehicle treated group than in the PD325901 treated group (Independent-samples t-test: t17 = 7.896, p<0.0001), indicating that PD325901 injection partially perturbed the recognition of the familiar object. (I) Total time of object exploration was equivalent between groups, meaning that PD325901 injection had no influence on the motivational state of the animals (Independent-samples t-test: t17 = 0.857, p=0.403). (J) The D.I. of the PD325901-injected group was lower, by 23.24%, indicating that 25 mg/kg of PD325901 significantly interfered with the retrieval of recognition memory (PD325901: 0.087 ± 0.027, VEH: 0.376 ± 0.025; Independent-samples t-test: t17 = −7.898, p<0.0001), although still above chance level (One-sample t-test, PD325901: t9 = 3.283, p=0.009; VEH: t8 = 15.264, p<0.0001). (K) Analysis of the R.I. showed that the percentage of time spent exploring the familiar object significantly decreased compared with the one

*Figure 11 continued on next page*

*Figure 11 continued*

observed during the first exposition to the same object for both groups (Paired-samples t-test, PD: $t9 = 3.478$, p<0.01; VEH: $t8 = 10.260$, p<0.0001), but the level of recognition memory in the PD325901 treated group is strongly diminished, by 23.83% (PD325901: $0.093 \pm 0.027$, VEH: $0.390 \pm 0.027$; Independent-samples t-test: $t17 = -7.856$, p<0.0001), although still above chance level (One-sample t-test, PD: $t9 = 3.575$ p<0.01; VEH: $t8 = 14.672$, p<0.0001). PD325901 (n = 10), VEH (n =9). ▲▲▲▲ p<0.0001, ****p<0.0001. Data are shown as mean with SEM.

ERK activation in response to cocaine and, most importantly, capable of attenuating both Pavlovian and operant responding in response to cocaine in mice.

PD325901 is a highly selective allosteric MEK1/2 inhibitor that does not compete with ATP or ERK1/2. It is a particularly appealing drug since it shows a high potency with a low IC50 in vivo. Moreover, compared to its predecessor CI-1040, PD325901 exhibits a greater solubility, leading to improved bioavailability and metabolic stability (*Uehling and Harris, 2015*; *Wu and Park, 2015*). Consistent with this, PD325901 has demonstrated potent anticancer activity both in vitro, by inhibiting tumour cell proliferation, and in vivo, by suppressing tumour growth in a broad spectrum of xenograft models (*Solit et al., 2006*). Based on these promising preclinical studies, PD325901 has been subsequently advanced into clinical trials. The effectiveness of PD325901 in patients has been evaluated in a Phase I/II trial for the treatment of colon cancer, breast cancer and melanoma (NCT00147550, Pfizer), in a Phase II study for the treatment of non-small cell lung cancer (NCT00174369, Pfizer), and in another Phase II study in one previously-treated patient with advanced non-small cell lung cancer (NCT02297802, Sharp Healthcare). In addition, several combination therapies using PD325901 have been evaluated in order to expand its clinical application (NCT01347866, Pfizer; Phase I, NCT02022982, Dana-Farber Cancer Institute; Phase I/II, NCT02039336, The Netherlands Cancer Institute; Phase I, NCT02510001, University of Oxford; NCT020096471, University of Alabama at Birmingham).

Notably, the first completed analysis of a phase II study of PD325901 for cancer therapy showed some neurological adverse effects, including most commonly visual disturbances, but also dizziness, hallucinations, and balance and gait disorders (*Haura et al., 2010*; *LoRusso et al., 2010*). Here, we speculate that these early clinical observations indirectly confirm the ability of PD325901 to penetrate the human brain and exert its anti Ras-ERK activity. Although the toxicity issue is of unquestionable relevance, the protocol used in clinical trials for cancer therapy consisted of repeated and intermittent oral administrations of the compound. In contrast, with our protocol we demonstrated a remarkable inhibitory effect on both CPP and SA with a single administration, suggesting that such a treatment in humans should not cause major complications. Importantly, the ability of a single dose of PD325901 to accelerate the extinction of cocaine self-administration is obviously a key finding, which for the first time supports the idea that a systemic Ras-ERK inhibition may lead to a significant reduction in behaviours mediated by cocaine-cue associations. Obviously, it remains to be seen whether such an effect could be replicated in patients and whether a single dose/behavioural treatment would be sufficient to block relapse. One important concern related to future clinical developments is the potential side effects of PD325901 on memory mechanisms. Indeed, our own data indicate that this drug may cause memory dysfunctions both during the acquisition and the retrieval phases, but at doses that are significantly higher (25 mg/kg) than those used in cancer therapy and in our cocaine experiments. Importantly, the fact that we propose here to administer only one or a few doses of PD325901 at 10 mg/kg to treat cocaine addiction should be reassuring since such protocols are unlikely to cause permanent memory impairments. In addition, we would like to note that in our cocaine behavioural paradigms we used a dose which is far above the minimal (2.5 mg/kg) effective one in blocking ERK phosphorylation and thus it is likely that doses lower than 10 mg/kg could still be of therapeutic value.

A comparison with the only blood-brain barrier penetrating Ras-ERK inhibitor previously available, SL327, indicates that PD325901 may offer some significant advantages. It is a classical MEK inhibitor, and in that sense its action may be qualitatively similar to SL327, nevertheless it is much more potent (almost three orders of magnitude). Furthermore, in contrast to PD325901, SL327 at the dose necessary to completely block ERK phosphorylation (50 mg/kg) and commonly used in several behavioural paradigms, showed significant detrimental consequences on locomotion, suggesting potential off

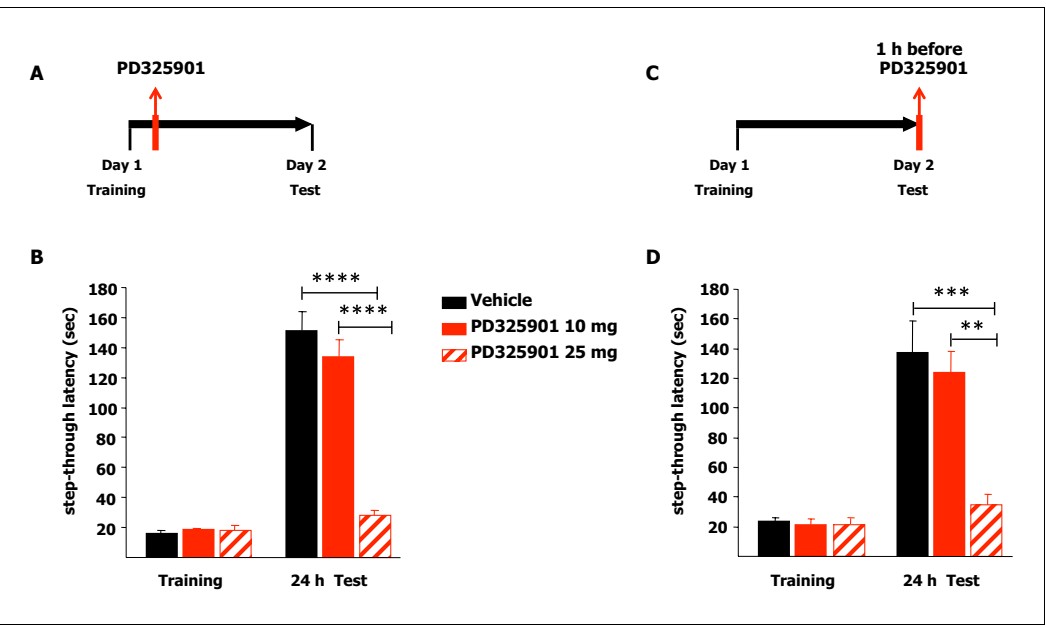

**Figure 12.** A single dose of PD325901 at 10 mg/kg does not affect fear memory in the Inhibitory avoidance test. (A) Animals were placed in a lighted arena and when they entered the dark compartment, they received an aversive footshock (0.2 mA × 2"). Immediately after training, animals received an injection of PD325901 (10 or 25 mg/kg, i.p.) or vehicle and were then evaluated for their memory on the subsequent day. (B) 10 mg/kg of PD325901 spared the consolidation of the inhibitory-avoidance learning. Before PD325901 treatment all groups showed equivalent latency to enter the dark compartment (One-way ANOVA, $F_{2,43}$ = 0.557, p=0.577). PD325901 treatment given immediately after training had a significant effect on memory consolidation on day 2 (Two-way ANOVA, treatment effect $F_{2,43}$ = 37.848, p<0.0001). In particular, only the higher dose of PD325901 disrupted consolidation when compared with VEH (Bonferroni's post-hoc, PD325901 10 mg vs VEH: p=0.667; PD325901 25 mg vs VEH: p<0.0001) and a dose-dependent effect was found for PD325901 (Bonferroni's post-hoc, PD325901 25 mg vs PD325901 10 mg: p<0.0001). Mice treated with 10 mg/kg of PD325901 and vehicle treated mice both showed significantly increased latency 24 hr after training (Paired-samples t-test, PD325901 10 mg: t15 = −9.695, p<0.0001; VEH: t15 = −11.129, p<0.0001), whereas mice injected with 25 mg/kg of PD325901 did not (Paired-samples t-test, PD325901 25 mg: t13 = −2.569, p=0.068), showing impaired memory consolidation. VEH (n =16), PD325901 10 mg (n = 16), PD325901 25 mg (n = 14). (C) Animals were placed in a lighted arena with free access to a dark compartment. When they entered the dark compartment, they received an aversive footshock (0.2 mA × 2"). On the subsequent day, one hour before the test session mice received an injection of PD325901 (10 or 25 mg/kg, i.p.) or vehicle and were then evaluated for their inhibitory-avoidance learning. (D) 10 mg/kg of PD325901 spared the retrieval of the inhibitory-avoidance learning. Before PD325901 treatment all groups showed equivalent latency to enter the dark compartment (One-way ANOVA, $F_{2,32}$ = 0.092, p=0.913). On day 2, PD325901 treatment altered the memory performance (Two-way ANOVA, treatment effect: $F_{2,32}$ = 9.536, p<0.001). Specifically, only 25 mg/kg of PD325901 significantly compromised memory retrieval when compared with VEH (Bonferroni's post-hoc, PD325901 10 mg vs VEH: p=1.000; PD325901 25 mg vs VEH: p<0.001) and a dose-dependent effect was found for PD325901 (Bonferroni's post-hoc, PD325901 25 mg vs PD325901 10 mg: p<0.01). Mice treated with 10 mg/kg of PD325901 and vehicle treated mice both showed significantly increased latency 24 hr after training (Paired-samples t-test, PD325901 10 mg: t10 = −7.493, p<0.0001; VEH: t13 = −4.804, p<0.0001), while mice injected with 25 mg/kg of PD325901 did not (Paired-samples t-test, PD325901 25 mg: t9 = −1.418, p=0.190), demonstrating a significantly impaired memory. VEH (n = 14), PD325901 10 mg (n = 11), PD325901 25 mg (n = 14). **p<0.01, ***p<0.001, ****p<0.0001. Data are shown as mean with SEM.

target effects that should be further evaluated (*Valjent et al., 2006b*). Most importantly, PD325901 has already been demonstrated to manifest an acceptable toxicity in humans and thus could be rapidly repositioned for the treatment of cocaine addiction.

Similarly, the two RB1 and RB3 peptides interfered with the expression of cocaine-induced place preference, albeit to a lesser extent. These findings suggest that these peptides are also capable of inhibiting cocaine-mediated responses. Considering that several other cell-penetrating peptides are

already in clinical trials for various conditions, including neurological disorders, our peptides may be in the near future further validated for clinical purposes (*Mehta et al., 2011*; *Hill et al., 2012*; *Warso et al., 2013*). Also, in comparison to SL327, these peptides may offer some advantages. While RB1 and RB3 show an IC50 toward ERK comparable to SL327, they posses an excellent water solubility (SL327 is dissolved in organic solvents) and potentially lower toxicity. In addition, RB1 and RB3 act on different Ras-ERK components (ERK1/2- and Ras- interacting partners, respectively) than SL327, thus expanding the portfolio of potential targets.

Finally, beyond the field of drug addiction, these drugs may also be relevant for the experimental treatment of other brain disorders, most notably a class of neurodevelopmental disorders collectively named RASopathies, which are characterised by activating mutations in the Ras pathway. Indeed, it has been shown that the use of Ras-ERK inhibitors can reverse some of the associated symptoms in animal models (*Lee et al., 2014*; *Papale et al., 2016*). Thus, further development of the tools outlined here may not only facilitate the investigation of the neurobiological bases of drug addiction, but may possibly lead to the development of pharmacotherapies for the treatment of other neuropsychiatric conditions.

## Materials and methods

### Animals and drugs

Male C57Bl/6 mice (Charles River) were single-housed in a temperature-controlled (21°C) environment maintained on a 12-hr light-dark cycle (lights on at 7 a.m.). Food and water were available ad libitum. All behavioural testing was conducted during the light phase between 08.00 hr and 17.00 hr. All procedures involving animals were conducted according to the European Community guidelines (*Directive 2010/63/EU)* and previously approved by the Institutional Animal Care and Use Committee (IACUC) protocols (# 585) of the IRCCS-San Raffaele Scientific Institute and the Regierungspräsidium for Baden-Württemberg in Karlsruhe, Germany (#35–9185.81/G-49/16).

Every attempt was made to ensure minimum discomfort to the animals at all times.

RB1 was designed around the 59–73 portion, the KIM sequence (*Liu et al., 2006*), of the ERK1/2 specific phosphatase MKP3. RB3 was designed, using the MOE software package, (Molecular Operating Environment, version 10.10, Chemical Computing group, Montreal Canada, http://www.chemcomp.com) by aligning and superposing the CDC25 domain of Ras-GRF1 (*Freedman et al., 2006*) the published crystal structure of a ternary Ras:SOS:Ras*GDP complex (*Sondermann et al., 2004*) using the default settings. The interacting surface between the two proteins was then visually analysed and the portion of the CDC25 domain between residues 1173 to 1203 was selected for the preparation of the final peptide.

RB1 (MGRKKRRQRRRPPQAPGIMLRRLQKGNLPVSRYPYDVPD), SCR RB1 (MGRKKRRQRRRPPQALSLKRLRSRGMNRTSATQSRYPYD), RB3 (GRKKRRQRRRPPCVPYLGMYLTDLVFIEEGTPNYTEDGLVN) and SCR RB3 (GRKKRRQRRRPPCFEVYPDSGDYTYEGELNGTLMVVPTN) were custom synthesised by GENECUST EUROPE (Luxembourg).

For all in vitro and in vivo experiments, batches of 200 mg, highly purified by high-performance liquid chromatography (HPLC) ($\geq$ 95% ) with C-terminal aminoacid (last) in D form and acetylated N-Terminal (first) aminoacid were used. For in vivo experiments the peptides were dissolved in PBS 1X and injected 20 mg/kg 1 hr before the cocaine injection.

Dabrafenib, Trametinib, Selumetinib (Selleck Chemicals) and PD325901 (Sigma Aldrich, UK) were dissolved in DMSO (20% final solution) and injected at the doses indicated. Cocaine hydrochloride (Sigma-Aldrich, Germany) was dissolved in physiological saline (0.9% NaCl) for intravenous (i.v.) infusion of 0.50 mg/kg in a 14 µl infusion or injected intraperitoneally (i.p.) at a dose of 20 or 25 mg/kg. L-Glutamic acid (Sigma Aldrich) was dissolved in sterile water.

### Ex vivo system in acute brain slices

Adult mice were decapitated after cervical dislocation and brain slices were freshly prepared according to the protocol described in (*Marti et al., 2012*; *Orellana et al., 2012*). The brains were rapidly removed from the skull and put on a cool glass plate filled with ice-cold sucrose-based dissecting solution (87 mM NaCl, 2.5 mM KCl, 7 mM $MgCl_2$, 1 mM $NaH_2PO_4$, 75 mM sucrose, 25 mM $NaHCO_3$, 10 mM D -glucose, 0.5 mM $CaCl_2$, 2 mM kynurenic acid), oxygenated with 95% $O_2$ and 5% $CO_2$, and

subsequently mounted on the vibratome stage (Vibratome, VT1000S-Leica Microsystems). 200 $\mu$m-thick slices were cut and transferred into the brain slice chamber (Brain slice chamber-BSC1 - Scientific System design Inc., Mississauga, ON, Canada) and allowed to recover for 1 hr at 32°C, with a constant perfusion of carboxygenated artificial cerebrospinal fluid (ACSF) in the presence of different doses of the cell penetrating peptides, PD325901 or their controls. 100 $\mu$M glutamate was applied to stimulate brain slices into the chamber for 10 min. After a rapid fixation in 4% PFA for 15 min at room temperature, slices were rinsed and cryoprotected overnight at 4°C in sucrose solution (30%). On the following day, slices were further cut into thinner slices of 18 $\mu$m using a cryostat (Leica CM1850) and mounted onto SuperFrost Plus slides (Thermo Scientific), and stored at $-20$°C until processing for immunohistochemistry or immunofluorescence.

## In vivo administration of drugs

At the indicated times after drug treatments, animals were anaesthetised and transcardially perfused with ice-cold buffered 4% PFA.

Brains were extracted, post-fixed overnight and transferred to 30% buffered sucrose for 24 hr. Coronal sections were cut to a 35 μm thickness on a freezing microtome and stored in a cryoprotective solution at $-20$°C until processing for immunohistochemistry or immunofluorescence.

## Immunohistochemistry

Immunohistochemistry was performed following the protocol described in (*Fasano et al., 2009*; *Bido et al., 2015*). 1 hr after blocking in 5% normal goat serum and 0.1% Triton X-100 solution, slices were incubated overnight at 4°C with anti-phospho-p44/42 MAP kinase (Thr202/Tyr204) (1:200, Cell Signaling Technology Cat# 4370L, RRID: AB_2297462). Sections were then incubated with biotinylated goat anti-rabbit IgG (1:200, Vector Laboratories Cat# BA-1000, RRID: AB_2313606) for 2 hr. Detection of the bound antibodies was carried out using a standard peroxidase-based method (ABC-kit, Vectastain, Vector Labs), followed by a DAB and $H_2O_2$ solution. Images were acquired from the dorsal striatum (for ex-vivo experiments) and ventral striatum (for in vivo experiments) using a Zeiss Axioplan 2 microscope under a 20X objective.

## Immunofluorescence

Immunofluorescence was performed as described in (*Marti et al., 2012*; *Orellana et al., 2012*; *Bido et al., 2015*). 1 hr after blocking in 5% normal goat serum and 0.1% Triton X-100 solution, slices were incubated overnight at 4°C with one of the following primary antibodies: anti-phospho-S6 ribosomal protein (Ser235/236) (1:200, Cell Signaling Technology Cat# 2211, RRID: AB_331679) or anti-phospho (Ser10)-acetylated (Lys14) histone H3 (1:1000, Millipore Cat# 07–081, RRID: AB_310366) and anti-NeuN (1:1000, Millipore Cat# MAB377B, RRID: AB_177621). Sections were then incubated for 1 hr at room temperature with the following secondary antibodies: Alexa Fluor 546 conjugated anti-mouse (1:200, Thermo Fisher Scientific Cat# A-11003, RRID: AB_2534071) and Alexa Fluor 488 conjugated anti-rabbit (1:500, Thermo Fisher Scientific Cat# A-11008, RRID: AB_143165). Single and double-labelled images (1024 $\times$ 1024 $\mu$m) were obtained at 40X magnification from striatum using a laser scanning confocal microscopy (Leica SP2) equipped with the corresponding lasers and appropriate filters sets to avoid the cross-talk between the fluorochromes.

## Image quantification and statistics

Neuronal quantification was performed using ImageJ software.

For ex-vivo immunohistochemistry experiments, the number of phospho-ERK positive cells was counted in 3 consecutive sections for each treatment in 2 fields per section. The level of activation was expressed as arbitrary units (AU). AU=% p-cells in glu$^+$ scramble treated samples -% p-cells in glu$^-$ scramble treated samples.

The IC50 was calculated for the specific response using GraphPad Prism software.

For in vivo immunohistochemistry experiments, the number of phospho-ERK positive cells was counted in 2 consecutive rostral sections per mouse in 2 fields per section.

In ex-vivo immunofluorescence experiments, the number of phospho-S6 or phospho-Ac-H3 positive cells among NeuN positive cells was counted in 3 consecutive sections for each treatment in 3 fields per section. The level of activation and the IC50 were calculated as described above.

In in-vivo immunofluorescence experiments, the number of phospho-S6 or -Ac-H3 immunoreactive neurons among NeuN positive neurons was counted in 2 consecutive rostral sections per mouse in 4 fields per section.

Comparisons between groups of different treatments were performed using two-way ANOVA and post-hoc analysis with Bonferroni's post-hoc test, in GraphPad Prism 5 software.

## Cocaine conditioning place preference (CPP)

Conditioned place preference testing was performed as previously described (*Fasano et al., 2009*). Briefly, CPP was performed in a three-chamber apparatus containing two large compartments with distinct patterns on the floors and walls, separated by a central neutral area. During the pre-conditioning phase, mice were allowed 18 min free access to the apparatus. Only mice that spent less than 75% of their time in any one compartment were included in the study. During the conditioning phase (days 2–7), mice were confined to one compartment for 20 min after an injection of cocaine (20 mg/kg, i.p.) on days 2, 4, and 6, or to the other compartment after a saline injection on days 3, 5, and 7. Saline-treated mice received six pairings of saline. One hour prior to the conditioning test (day 8, test 1), mice were treated with vehicle or inhibitors (PD325901, 10 mg/kg, RB1/RB3 mix, 20 mg/kg or Trametinib, 10 mg/kg) and then allowed 18 min free access to the apparatus. A place preference score was calculated for each mouse as the difference between pre-conditioning and post-conditioning time spent in the drug-paired compartment. After 2 weeks (day 22, test 2), mice were retested in the CPP apparatus in a drug-free state. The day following test 2 (day 23), mice underwent one conditioning session (20 min) in the CPP apparatus immediately after cocaine or saline injection (20 mg/kg, i.p.). CPP was tested the following day (day 24, test 3).

## Novel object recognition (NOR)

The novel object recognition (NOR) test was performed in a square arena of 40 × 40 cm. On day 1, mice were first allowed to habituate to the testing arena in a 5 min session. Thigmotaxis (i.e. the percentage of time spent near the walls) was measured as an index of anxiety by analysing the visual recording made with a video camera. On day 2, animals underwent the training phase (10 min), in which two identical objects were introduced into the arena before allowing the mouse to explore. The amount of time that the rodents spent exploring each object was scored. Finally, on day 3, mice were tested for their 24 hr memory (10 min). The discrimination index (DI), defined as the difference between the exploration time for the novel object and the one for the familiar object, divided by total exploration time, was calculated. The recognition index (RI) measures the ability of an animal to recognise a same object at different time points and it is based on the fact that if an animal maintains the memory of the previously encountered familiar object, its percentage of exploration for that object will decrease when the animal is exposed to the same object for a second time. RI was computed as follows: [(percentage of exploration of the undisplaced familiar object during the training) – (percentage of exploration of the undisplaced familiar object during the test)] / percentage of exploration of the undisplaced familiar object during the training (*d'Isa et al., 2014*).

Two doses of PD325901, 10 and 25 mg/kg were given either at day 2 to interfere with the acquisition process of recognition memory or at day 3 to perturb the retrieval of that memory.

The sessions were recorded with the video tracking software SMART (Panlab, Barcelona, Spain).

## Inhibitory avoidance

The apparatus was divided into two compartments: a larger one made of white Plexiglas, kept strongly illuminated and fitted at one end with a guillotine door, and a smaller one made of black Plexiglas (Panlab, Barcelona, Spain). A floor grid placed in the dark chamber received electrical current. On the training day each mouse was placed in the lit compartment, facing away from the dark compartment. Mice show a natural tendency to leave a well-lit chamber for a darkened one, and they do so after a short latency. When a mouse entered the dark compartment with all four paws 1 footshock (0.2 mA x 2") was delivered. The time spent in the lit side (step-through latency) 24 hr later is the measure of 'inhibitory avoidance'. A maximum step-through latency of 360 s was allowed in the test session (*Brambilla et al., 1997*).

Two doses of PD325901, 10 and 25 mg/kg, were given either at day 1 after training to interfere with the consolidation of fear memory or at day 2 to perturb the retrieval of that memory.

## Locomotor activity

Horizontal activity was recorded in locomotor activity boxes equipped with photocell beams (Ugo Basile, Comerio, Italy) during a single session starting 30 min after PD325901 injection (10 mg/kg, i. p.) and recorded every 5 min for a total of 6 trials. At the end of this session mice received a cocaine (20 mg/kg, i.p.) challenge and the effect of PD325901 pre-treatment on the hyper-locomotion induced by cocaine was evaluated for the subsequent 30 min (*Fasano et al., 2009*).

## Cocaine self-administration (SA)

SA was assessed in 12 operant chambers (Med Associates, USA) housed in light- and sound-attenuating cubicles. Each chamber (24.1 × 20.3 × 18.4 cm) is equipped with two levers and a drug delivery system connected via infusion pump (PHM-100, Med-Associates, USA) located outside the cubicle. Operant chambers were controlled using Med-PC IV (Med Associates, USA) software. Mice were implanted with an indwelling intravenous catheter (made in-house) into the jugular vein. Catheter patency was maintained with 0.15 ml heparanised saline (100 i.u./ml) containing Baytril (0.7 mg/ ml) administered daily throughout the experiment. After 3-day recovery, self-administration in mice began.

For all behavioural experiments, statistical analyses were conducted using SPSS software (Stat-Soft, USA). Significance was set at $p < 0.05$.

## Acknowledgements

We thank Dr Raffaele d'Isa and Dr Nicola Solari for helpful comments and discussion on the experimental design. Molecular graphics images were produced using the UCSF Chimera package from the Computer Graphics Laboratory, University of California, San Francisco (supported by NIH P41 RR-01081).

## Additional information

### Funding

| Funder | Grant reference number | Author |
| --- | --- | --- |
| Deutsche Forschungsgemeinschaft | 383/5-1 | Rainer Spanagel |
| ERANET | COCADDICT | Rainer Spanagel |
| Fondazione Cariplo | | Riccardo Brambilla |
| Michael J. Fox Foundation for Parkinson's Research | | Riccardo Brambilla |
| Parkinson's UK | | Riccardo Brambilla |
| Ministero della Salute | | Riccardo Brambilla |
| Compagnia di San Paolo | | Riccardo Brambilla |

The funders had no role in study design, data collection and interpretation, or the decision to submit the work for publication.

### Author contributions

AP, IMM, REB, RS, RB, SF, Conception and design, Acquisition of data, Analysis and interpretation of data, Drafting or revising the article; MTI, LM, FM, Acquisition of data, Analysis and interpretation of data; AB, Conception and design, Drafting or revising the article

### Author ORCIDs

Alessandro Papale, http://orcid.org/0000-0002-8794-0171
Rainer Spanagel, http://orcid.org/0000-0003-2151-4521
Riccardo Brambilla, http://orcid.org/0000-0003-3569-5706
Stefania Fasano, http://orcid.org/0000-0002-3696-7139

## Ethics

Animal experimentation: All procedures involving animals were conducted according to the European Community guidelines (Directive 2010/63/EU) and previously approved by the Institutional Animal Care and Use Committee (IACUC) protocols (# 585) of the IRCCS-San Raffaele Scientific Institute and the Regierungspräsidium for Baden-Württemberg in Karlsruhe, Germany (#35-9185.81/ G-49/16). Every attempt was made to ensure minimum discomfort to the animals at all times.

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
