## [Decision Letter]

Thank you for submitting your article "Impairment of cocaine-mediated behaviors by clinically relevant Ras-ERK inhibitors" for consideration by *eLife*. Your article has been reviewed by three peer reviewers, including Claes Wahlestedt (Reviewer #3), and a member of our Board of Reviewing Editors, and the evaluation has been overseen by a Senior Editor.

The reviewers have discussed the reviews with one another and the Reviewing Editor has drafted this decision to help you prepare a revised submission.

Summary:

This study provides interesting information concerning the potential clinical use of drugs acting as inhibitors of the extracellular signal regulated kinase (ERK) cascade for brain diseases including cocaine addiction. Extensive published work already indicates the critical role played by ERK in a multitude of physio-pathological processes, including learning, memory and drug addiction. One MEK inhibitor, PD325901, is already in clinical trials for cancer, thus it has potential be used as a treatment for brain disorders in humans. The idea of utilizing systemically administrable ERK inhibitors to reverse addictive-like behaviors is intriguing, and here the authors test two brain-penetrating peptides and a small molecule compound, which they show inhibit ERK signalling in the brain and reduce cocaine conditioned place preference. Most importantly they report that a single dose of PD325901 enhances extinction of drug seeking in a self-administration procedure that is highly relevant as a clinical model.

Essential revisions:

All three reviewers have two major concerns, which we consider addressable in the context of the current study.

1) It is essential to provide control experiments that test whether a single dose of PD325901 in the context of the cocaine studies is associated with a general cognitive deterioration and/or general effects on locomotor function. Such non-specific effects on learning and memory or task performance would diminish the utility of PD325901 for clinical purposes. These cognitive data could be provided using a relatively simple paradigm (e.g. novel object recognition, passive avoidance, fear conditioning, conditioned place aversion) so long as it would reveal whether the MAPK-blocking drug treatments used in this study selectively attenuate drug-related memories or generally inhibit all learning and memory once given. In addition, since the authors show that three compounds that act as MEK inhibitors – Trametinib, Selumetinib, and Dabrafenib – do not reduce ERK phosphorylation in the brain, it would be appropriate to test at least one of them as a negative control in one behavioral test (e.g. CPP).

2) The authors need to improve and expand their discussion of the comparison of the drugs developed in this study (PD325901 and RB1/3) versus SL327 in the realm of neurological disorders. The primary difference between PD325901 and SL327 is the lower IC50 of the first. Though SL327 has not been tested in clinical trials it is used in experimental models at high doses thus the main conceptual difference between PD325901 and SL327 is of quantitative, rather than qualitative in nature. Especially since this manuscript does not explore mechanisms of the behavioral actions of ERK inhibition measured in this study, this discussion should comment on the possible differences, or lack of differences, with regard to the mechanisms of action of the two drugs.

---

## [Author Response]

*1) It is essential to provide control experiments that test whether a single dose of PD325901 in the context of the cocaine studies is associated with a general cognitive deterioration and/or general effects on locomotor function. Such non-specific effects on learning and memory or task performance would diminish the utility of PD325901 for clinical purposes. These cognitive data could be provided using a relatively simple paradigm (e.g. novel object recognition, passive avoidance, fear conditioning, conditioned place aversion) so long as it would reveal whether the MAPK-blocking drug treatments used in this study selectively attenuate drug-related memories or generally inhibit all learning and memory once given. In addition, since the authors show that three compounds that act as MEK inhibitors – Trametinib, Selumetinib, and Dabrafenib – do not reduce ERK phosphorylation in the brain, it would be appropriate to test at least one of them as a negative control in one behavioral test (e.g. CPP).*

We agree entirely with the comments made above. It is well known that inhibition of ERK signalling in a number of behavioural paradigm leads to cognitive impairments and this fact may reduce the clinical relevance of Ras-ERK inhibitors for treating addiction.

We have addressed the specific issues of the potential memory and locomotor impairments as follows:

1) We have analysed the distance travelled in the place preference apparatus during the pre-conditioning phase at day 1 and test session at day 8, i.e. 1 hour after PD injection and we have found a clear habituation effect to the apparatus with no difference in locomotion among groups (See Figure 8—figure supplement 1).

2) We have also included a study of locomotor function using the locomotor activity cages (equipped with infrared movement detector system), upon administration of a single dose of PD325901 (10 mg). We monitored PD325901 effects before and after a cocaine challenge for 1-hour time interval and we confirmed that at this dose no detrimental effects on locomotion can be detected (See Figure 8—figure supplement 2).

3) We have now provided compelling evidence that our selected dose of PD, i.e. 10 mg/kg, has no adverse consequences on memory performance in the Novel Object Recognition (NOR). Specifically, we applied two doses of PD325901 (10 and 25 mg/kg) in two distinct phases of the behavioural test, the acquisition and the retrieval processes, and we found that only the higher dose partially perturbed them. As an additional control on motor function, both doses did not influence the exploration of the objects in the arena (Figure 10 and Figure 11).

4) We have added another behavioural paradigm, the inhibitory avoidance, to verify any possible confounding memory effect in response to PD injection in an aversive condition. Two doses of PD325901 (10 and 25 mg/kg) were applied during consolidation and retrieval processes, and also in this case only the higher dose did disrupt fear memory (Figure 12).

5) We have tested Trametinib, a MEK inhibitor that does not pass the blood brain barrier, in the conditioned place preference, for comparison with the PD325901. As expected, Trametinib at 10 mg/kg, the same dose used for PD325901, did not reduce the rewarding properties of cocaine in CPP (Figure 8—figure supplement 3).

The new experimental evidence with NOR and Inhibitory avoidance tests strengthen our hypothesis that this potent MEK inhibitor is much more selective for the drug-related memories and does not compromise the general brain function at least at 10 mg/kg. We have obviously discussed the potential side effects at higher doses, highlighting the fact that our therapeutic approach is unlikely to cause permanent memory deficits since it only requires a single dose treatment.

*2) The authors need to improve and expand their discussion of the comparison of the drugs developed in this study (PD325901 and RB1/3) versus SL327 in the realm of neurological disorders. The primary difference between PD325901 and SL327 is the lower IC50 of the first. Though SL327 has not been tested in clinical trials it is used in experimental models at high doses thus the main conceptual difference between PD325901 and SL327 is of quantitative, rather than qualitative in nature. Especially since this manuscript does not explore mechanisms of the behavioral actions of ERK inhibition measured in this study, this discussion should comment on the possible differences, or lack of differences, with regard to the mechanisms of action of the two drugs.*

We have now extensively discussed the differences between our novel drugs and SL327.